# Equivariant Flow Matching with Hybrid Probability Transport for 3D Molecule Generation

**Yuxuan Song**[1,*]**, Jingjing Gong**[1,*]**, Minkai Xu**[2]**, Ziyao Cao**[1]
**Yanyan Lan** [1,3]**, Stefano Ermon**[2]**, Hao Zhou**[1]**,Wei-Ying Ma**[1]
[1] Institute of AI Industry Research(AIR), Tsinghua University
[2] Computer Science Department, Stanford University
[3]Beijing Academy of Artificial Intelligence
{songyuxuan,gongjingjing}@air.tsinghua.edu.cn

## Abstract

The generation of 3D molecules requires simultaneously deciding the categorical features (atom types) and continuous features (atom coordinates). Deep generative models, especially Diffusion Models (DMs), have demonstrated effectiveness in generating feature-rich geometries. However, existing DMs typically suffer from unstable probability dynamics with inefficient sampling speed. In this paper, we introduce geometric flow matching, which enjoys the advantages of both equivariant modeling and stabilized probability dynamics. More specifically, we propose a hybrid probability path where the coordinates probability path is regularized by an equivariant optimal transport, and the information between different modalities is aligned. Experimentally, the proposed method could consistently achieve better performance on multiple molecule generation benchmarks with $4.75\times$ speed up of sampling on average.[2]

## 1 Introduction

Geometric generative models aim to approximate the distribution of complex geometries and emerge as an important research direction in various scientific domains. A general formulation of the geometries in scientific fields could be the point clouds where each point is embedded in the Cartesian coordinates and labeled with rich features. For example, the molecules are the atomic graphs in 3D [44] and the proteins could be seen as the proximity spatial graphs [15]. Therefore, with the ability of density estimation and generating novel geometries, geometric generative models have appealing potentials in many important scientific discovery problems, *e.g.*, material science [36], de novo drug design [11] and protein engineering [45].

With the advancements of deep generative modeling, there has been a series of fruitful research progresses achieved in geometric generative modeling, especially molecular structures. For example, [9, 30] and [42] proposed data-driven methods to generate 3D molecules (in silico) with autoregressive and flow-based models respectively. However, despite great potential, the performance is indeed limited considering several important empirical evaluation metrics such as validity, stability, and molecule size, due to the insufficient capacity of the underlying generative models [40]. Most recently, diffusion models (DMs) have shown surprising results on many generative modeling tasks which generate new samples by simulating a stochastic differential equation (SDE) to transform the prior density to the data distribution. With the simple regression training objective, several attempts [13] on applying DMs in this field have also demonstrated superior performance. However, existing

---

*Equal Contribution, Hao Zhou (zhouhao@air.tsinghua.edu.cn) is the corresponding author.
[2]The code is available at https://github.com/AlgoMole/MolFM

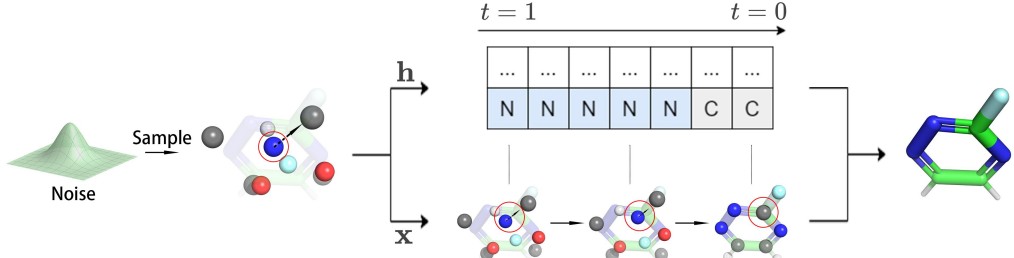

Figure 1: Illustration of EquiFM. We define a hybrid path for generating molecules $\mathbf{g} = \langle \mathbf{x}, \mathbf{h} \rangle$, where $\mathbf{x}$ is trained on an equivariant optimal transport path and $\mathbf{h}$ is trained on a path whose information quantity is aligned with $\mathbf{x}$'s path. The sampling is conducted by solving an ODE, *i.e.* $\mathbf{g}_0 = \text{ODESolve}(\mathbf{g}_1, v_\theta, 1, 0)$.

DM-based methods typically suffer from unstable probability dynamics which could lead to an inefficient sampling speed also limit the validity rate of generated molecules.

In this work, we propose a novel and principled flow-matching objective, termed Equivariant Flow-Matching (EquiFM), for geometric generative modeling. Our method is inspired by the recent advancement of flow matching [26], a simulation-free objective for training CNFs that has demonstrated appealing generation performance with stable training and efficient sampling. Nevertheless, designing suitable geometric flow-matching objectives for molecular generation is non-trivial:

(1) the 3D skeleton modeling is sensitive, *i.e.*, a slight difference in the atom coordinates could affect the formulation of some certain types of bonds; (2) the atomic feature space consists of various physical quantities which lies in the different data modality, *e.g.*, charge, atom types, and coordinates are correspondingly discrete, integer, and continuous variables. To this end, we highlight our key innovations as follows:

- For stabling the 3D skeleton modeling, we introduce an Equivariant Optimal-Transport to guide the generative probability path of atom coordinates. The improved objective implies an intuitive and well-motivated prior, *i.e.* minimizing the coordinates changes during generation, and helps both stabilize training and boost the generation performance.

- Towards the modality inconsistency issues, we proposed to differ the generative probability path of different components based on the information quantity and thus introduce a hybrid generative path. The hybrid-path techniques distinguish different modalities without adding extra modeling complexity or computational load.

- The proposed model lies in the scope of continuous normalizing flow, which is parameterized by an ODE. We can use an efficient ODE solver during the molecule generation process to improve the inference efficiency upon the SDE simulation required in DMs.

A unique advantage of EquiFM lies in the framework enriching the flexibility to choose different probability paths for different modalities. Besides, the framework is very general and could be easily extended to various downstream tasks. We conduct detailed evaluations of the EquiFM on multiple benchmarks, including both unconditional and property-conditioned molecule generation. Results demonstrate that EquiFM can consistently achieve superior generation performance on all the metrics, and $4.75\times$ speed up on average. Empirical studies also show a significant improvement in controllable generation. All the empirical results demonstrate that the EquiFM enjoys a significantly higher modeling capacity and inference efficiency.

## 2 Related Work

**Flow Matching and Diffusion Models** Diffusion models have been studied in various research works such as [47, 12, 49], and have recently shown success in fields like high-dimensional statistics [41], language modeling [23], and equivariant representations [13]. Loss-rescaling techniques for diffusion models have been introduced in [48], while enhancements to the architecture incorporating classifier guidance are discussed in [8]. Noise schedule learning techniques have also been proposed in [33, 18]. Diffusion models suffer from unstable probability dynamics and inefficient sampling, which limits

their effectiveness in some scenarios. Flow matching is a relatively new approach that has gained attention recently. Research works such as [26, 1, 28] have proposed this simulation-free objective for training continuous normalizing flow. It involves other probability paths besides the diffusion path and could potentially offer better sampling efficiency through ODE solving. Furthermore, the follow-ups [37, 50] proposed to use the OT couplings to straighten the marginal probability paths. However, the application of flow matching to geometric domains requires designing appropriate probability paths, which is an area that remains unexplored.

**3D Molecule Generation** Previous studies have primarily focused on generating molecules as 2D graphs [14, 27, 46], but there has been increasing interest in 3D molecule generation. G-Schnet and G-SphereNet [9, 30] have utilized autoregressive methods to construct molecules by sequentially attaching atoms or molecular fragments. These frameworks have also been extended to structure-based drug design [24, 35, 38]. However, this approach requires careful formulation of a complex action space and action ordering. Other approaches use atomic density grids that generate the entire molecule in a single step by producing a density over the voxelized 3D space [31]. Nevertheless, these density grids lack the desirable equivariance property and require a separate fitting algorithm.

In the past year, the attention has shifted towards using DMs for 3D molecule generation [13, 52, 53], with successful applications in target drug generation [25], antibody design [29], and protein design [2, 51]. However, our method is based on the flow matching objective and hence lies in a different model family, *i.e.* continuous normalizing flow, which fundamentally differs from this line of research in both training and generation.

## 3 Backgrounds

### 3.1 Flow Matching for Non-geometric Domains

In this section, we provide an overview of the general flow matching method to introduce the necessary notations and concepts based on [26]. The data distribution is defined as $q$, $x_0$ represents a data point from $q$ and $x_1$ represents a sample from the prior distribution $p_1$. The *time-dependent* probability path is defined as $p_{t\in[0,1]} : \mathbb{R}^d \to \mathbb{R}_{>0}$, and the time-dependent vector field is defined as $v_{t\in[0,1]} : \mathbb{R}^d \to \mathbb{R}^d$. The vector field uniquely defines time-dependent flow $\psi_{t\in[0,1]} : \mathbb{R}^d \to \mathbb{R}^d$ by the following ordinary differential equation (ODE):

$$\frac{\mathrm{d}}{\mathrm{d}t}\psi_t(x) = v_t(\psi_t(x)), \psi_1(x) = x \tag{1}$$

[4] proposed to train the parameterized flow model $\psi_t$ called a continuous normalizing flow (CNF) with black-box ODE solvers. Such a model could reshape a simple prior distribution $p_1$ to the complex real-world distribution $q$. CNFs are difficult to train due to the need for numerical ODE simulations. [26] introduced flow matching, a simulation-free objective, by regressing the neural network $v_\theta(x, t)$ to some target vector field $u_t(x)$:

$$\mathcal{L}_{\mathrm{FM}}(\theta) = \mathbb{E}_{t, p_t(x)} \|v_\theta(x, t) - u_t(x)\|^2 \tag{2}$$

The objective $\mathcal{L}_{\mathrm{FM}}$ requires access to the vector field $u_t(x)$ and the corresponding probability path $p_t(x)$. However, these entities are difficult to define in practice. Conversely, the conditional vector field $u_t(x \mid x_0)$ and the corresponding conditional probability path $p_t(x \mid x_0)$ are readily definable.

The probability path can be marginalized from a mixture of conditional probability path $p_t(x) = \int p_t(x \mid x_0)q(x_0)\mathrm{d}x_0$, and the vector field $u_t(x)$ can be marginalized from conditional vector field as $u_t(x) = \mathbb{E}_{x_0 \sim q} \frac{u_t(x|x_0)p_t(x|x_0)}{p_t(x)}$. This illustrates how $u_t(x)$ and $p_t(x)$ are related to their conditional form, and [26] further proved that with the conditional vector field $u_t(x \mid x_0)$ generating the conditional probability path $p_t(x \mid x_0)$, the marginal vector field $u_t(x)$ will generate the marginal probability path $p_t(x)$. The observation inspires the new conditional flow matching (CFM) objective:

$$\mathcal{L}_{\mathrm{CFM}}(\theta) = \mathbb{E}_{t, q(x_0), p_t(x|x_0)} \|v_\theta(x, t) - u_t(x \mid x_0)\|_2^2 \tag{3}$$

The CFM objective enjoys the tractability for optimization, and optimizing the CFM objective is equivalent to optimizing Eq. 2, *i.e.*, $\nabla_\theta \mathcal{L}_{\mathrm{FM}}(\theta) = \nabla_\theta \mathcal{L}_{\mathrm{CFM}}(\theta)$. For the inference phase, ODE solvers could be applied to solve the Eq. 1, *e.g.*, $x_0 = \mathrm{ODESolve}(x_1, v_\theta, 1, 0)$. In this paper, we consider using the Gaussian conditional probability path, which lies in the form of $p_t(x \mid x_0) = \mathcal{N}\left(x \mid \mu_t(x_0), \sigma_t(x_0)^2 I\right)$. We introduce two probability paths utilized in the following of paper:

**Conditional Optimal Transport Path**   With the prior distribution $p_1$ defined as a standard Gaussian distribution, the empirical data distribution $p_0\left(x \mid x_0\right)$ is approximated with a peaked Gaussian centered in $x_0$ with a small variance $\sigma_{\min}$ as $\mathcal{N}\left(x \mid x_0, \sigma_{\min}^2 I\right)$. The probability path is $p_t(x \mid x_0) = \mathcal{N}\left(x \mid (1-t)x_0, (\sigma_{\min} + (1-\sigma_{\min})t)^2 I\right)$ and the corresponding flow is $\psi_t(x) = (\sigma_{\min} + (1-\sigma_{\min})t)x + (1-t)x_0$. Then the vector field could be obtained by Eq. 1 as: $u_t(\psi_t(x) \mid x_0) = \frac{\mathrm{d}}{\mathrm{d}t}\psi_t(x) = -x_0 + (1-\sigma_{\min})x$.

Put the above terms into Eq. 3, the reparameterized objective is as:

$$\mathcal{L}_{\text{CFM}}^{\text{OT}}(\theta) = \mathbb{E}_{t,q(x_0),p_1(x_1)} \left\| v_\theta(\psi_t(x_1), t) - (-x_0 + (1-\sigma_{\min})x_1) \right\|^2 \tag{4}$$

Intuitively, the conditional optimal transport objective tends to learn the transformation direction from noise to data sample in a straight line which could hold appealing geometric properties.

**Variance Preserving Path**   The variance-preserving (VP) path is defined as $p_t(x \mid x_0) = \mathcal{N}\left(x \mid \alpha_t x_0, (1-\alpha_t^2)I\right)$, where $\alpha_t = e^{-\frac{1}{2}T(t)}$ and $T(t) = \int_0^t \beta(s)ds$. Here $\beta$ is some noise schedule function. Following the Theorem. 3 in [26], the target conditional vector field of VP path could be derived as $u_t(x \mid x_0) = \frac{\alpha_t'}{1-\alpha_t^2}(\alpha_t x - x_0)$. $\alpha_t'$ denotes the derivative with respect to time. And the objective for VP conditional flow matching is as:

$$\mathcal{L}_{\text{CFM}}^{\text{VP}}(\theta) = \mathbb{E}_{t,q(x_0),p_t(x|x_0)} \left\| v_\theta(x, t) - \frac{\alpha_t'}{1-\alpha_t^2}(\alpha_t x - x_0) \right\|^2 \tag{5}$$

The VP path is flexible to control the information dynamics, *e.g.* correlation changes towards $x_0$ on the conditional probability path, by selecting different noise schedule functions.

## 4   Methodology

In this section, we formally describe the Equivariant Flow Matching (EquiFM) framework. The proposed method is inspired by the appealing properties of recent advancements in flow matching [26], but designing suitable probability paths and objectives for the molecular generation is however challenging [13]. We address the challenges by specifying a hybrid probability path with equivariant flow matching. The overall framework is introduced in Section. 4.1. And then we elaborate on the design details of the hybrid probability path of the equivariant variable and invariant variable in Section. 4.2 and Section. 4.3 respectively. A high-level schematic is provided in Figure. 1.

### 4.1   Equivariant Flow Matching

Recall molecule could be presented as the tuple $\mathbf{g} = \langle \mathbf{x}, \mathbf{h} \rangle$, where $\mathbf{x} = (\mathbf{x}^1, \dots, \mathbf{x}^N) \in X$ is the atom coordinates matrix and $\mathbf{h} = (\mathbf{h}^1, \dots, \mathbf{h}^N) \in \mathbb{R}^{N \times d}$ is the node feature matrix, such as atomic type and charges. Here $X = \left\{ \mathbf{x} \in \mathbb{R}^{N \times 3} : \frac{1}{N}\sum_{i=1}^N \mathbf{x}^i = \mathbf{0} \right\}$ is the Zero Center-of-Mass (Zero CoM) space, which means the average of the $N$ elements should be $\mathbf{0}$. We introduce the general form of equivariant flow matching in the following.

**SE(3) Invariant Probability Path**   For modeling the density function in the geometric domains, it is important to make the likelihood function invariant to the rotation and translation transformations. We could always make the probability path of equivariant variable $\mathbf{x}$ invariant to the translation by setting the prior distribution and vector field in the Zero CoM space, *i.e.* $\frac{1}{N}\sum_{i=1}^N v(x, t)^i = 0$. Formally, the rotational invariance could be satisfied by making the parameterized vector field equivariant and the prior $p_1$ invariant to the rotational transformations as shown in the following statement:

**Theorem 4.1.** *Let $(v_\theta^{\mathbf{x}}(\mathbf{g}, t), v_\theta^{\mathbf{h}}(\mathbf{g}, t)) = v_\theta(\mathbf{g}, t)$, where $v_\theta^{\mathbf{x}}(\mathbf{g}, t)$ and $v_\theta^{\mathbf{h}}(\mathbf{g}, t)$ are the parameterized vector field for $\mathbf{x}$ and $\mathbf{h}$. If the vector field is equivariant to any rotational transformation $\mathbf{R}$, i.e., $v_\theta(\langle \mathbf{Rx}, \mathbf{h} \rangle, t) = (\mathbf{R}(v_\theta^{\mathbf{x}}(\mathbf{g}, t)), v_\theta^{\mathbf{h}}(\mathbf{g}, t))$. With an rotational invariant prior function $p_1(\mathbf{x}, \mathbf{h})$, i.e., $p_1(\mathbf{Rx}, \mathbf{h}) = p_1(\mathbf{x}, \mathbf{h})$, then the probability path $p_{\theta,t}$ generated by the vector field $v_\theta(\cdot)$ is also rotational invariant.*

To make the vector field satisfy the equivariance constraint, we parameterize it with an Equivariant Graph Neural Network (EGNN) [43]. And more details can be found in Appendix C.

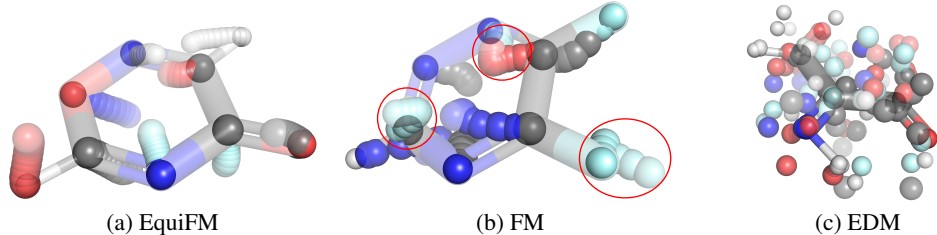

|     |     |     |
| --- | --- | --- |
| (a) EquiFM | (b) FM | (c) EDM |

Figure 2: Generation route visualization of different models. Note that a lighter color indicates an earlier step of an atom and a denser color corresponds to a later step. A change of base color indicates a change of atom type. EquiFM generates molecules in a straightforward route as shown in 2(a). Vanilla flow matching method 2(b) on the other hand, takes a detour while generating molecules, resulting in a route inward then outward before converging to a molecule. The generation process in EDM 2(c) is rather chaotic until the last few steps before converging into a molecule.

**Hybrid Probability Modeling**  We refer to the target conditional vector field on each part as $u_t^{\mathbf{x}}(\mathbf{g} \mid \mathbf{g}_0)$ and $u_t^{\mathbf{h}}(\mathbf{g} \mid \mathbf{g}_0)$ correspondingly, then we could get the objective in the following formulation:

$$\mathcal{L}_{\text{CFM}}(\theta) = \mathbb{E}_{t,q(\mathbf{g}_0),p_t(\mathbf{g}|\mathbf{g}_0)}[\|v_\theta^{\mathbf{x}}(\mathbf{g},t) - u_t^{\mathbf{x}}(\mathbf{g} \mid \mathbf{g}_0)\|_2^2 + \|v_\theta^{\mathbf{h}}(\mathbf{g},t) - u_t^{\mathbf{h}}(\mathbf{g} \mid \mathbf{g}_0)\|_2^2] \quad (6)$$

**Proposition 4.2.** *There could be joint probability path $p_t(\mathbf{g}|\mathbf{g}_0)$ which satisfies that $p_t(\mathbf{g}|\mathbf{g}_0) = p_t(\mathbf{x}|\mathbf{x}_0)p_t(\mathbf{h}|\mathbf{h}_0)$, and the conditional vector field on $\mathbf{x}$ and $\mathbf{h}$ is independent: $u_t^{\mathbf{x}}(\mathbf{g} \mid \mathbf{g}_0) = u_t(\mathbf{x} \mid \mathbf{x}_0), u_t^{\mathbf{h}}(\mathbf{g} \mid \mathbf{g}_0) = u_t(\mathbf{h} \mid \mathbf{h}_0)$.*

The above Proposition 4.2 states a special property of conditional flow matching, *i.e.*, in the multi-variable setting the probability path of each variable could be designed independently. Such property is appealing in our setting, as $\mathbf{x}$ and $\mathbf{h}$ hold different data types and come from different manifolds, thus it is intuitive to use different probability paths for modeling and generating the two variables.

## 4.2  Coordinates Matching with Equivariant Optimal Transport

We focus on the generation of coordinates variable $\mathbf{x}$. The conditional OT path (Eq. 4) could be a promising candidate as it tends to move the atom coordinates directly towards the ground truth atom coordinates along a straight line. However, directly applying the objective could be problematic in 3D molecule generation. With $\mathbf{x}_0$ as the point cloud from molecule distribution and $\mathbf{x}_1$ from the prior distribution, the objective in Eq. 6 tends to move the atom based on a random alignment between the atoms. Optimizing the vector field toward such a direction could involve extra variance for training and lead to a twisted and unstable generation procedure as shown in molecule generation visualization Fig. 2(b) and Fig. 2(c).

To address the above-mentioned issue, we first introduce the concept of equivariant optimal transport (EOT) between two geometries as follows:

**Definition 4.3.** Given two point clouds, $\mathbf{z} = (\mathbf{z}^1, \ldots, \mathbf{z}^N) \in \mathbb{R}^{N \times 3}$ and $\mathbf{y} = (\mathbf{y}^1, \ldots, \mathbf{y}^N) \in \mathbb{R}^{N \times 3}$. We define the equivariant optimal transport plan as

$$\pi^*, \mathbf{R}^* = \underset{\pi,\mathbf{R}}{\operatorname{argmin}} \|\pi(\mathbf{R}\mathbf{z}^1, \mathbf{R}\mathbf{z}^2, \ldots, \mathbf{R}\mathbf{z}^N) - (\mathbf{y}^1, \mathbf{y}^2, \ldots, \mathbf{y}^N)\|_2 \quad (7)$$

Here $\pi$ is a permutation of $N$ elements and $\mathbf{R} \in \mathbb{R}^{3 \times 3}$ stands for a rotation matrix in the 3D space. With and lie in the zero of mass space, the mappings in Eq. 7 are optimal towards any E(3) equivariant operations on either side of the point clouds. Therefore, the mappings are referred to as equivariant optimal transport(EOT).

The equivariant optimal transport finds the minimum straight-line distance between the paired atom coordinates upon all the possible rotations and alignment. We could then build a probability path based on the EOT map which could minimize the movement distance of atom coordinates for the transformation between the molecule data from $p_0$ and a sampled point cloud from $p_1$ as:

$$p_t = [\psi_t^{\text{EOT}}]_* p_1, \text{ where } \psi_t^{\text{EOT}}(\mathbf{x}) = (\sigma_{\min} + (1 - \sigma_{\min})t)\pi^*(\mathbf{R}^*\mathbf{x}) + (1 - t)\mathbf{x}_0 \quad (8)$$

**Proposition 4.4.** *The probability path implied by the EOT map, i.e. Eq. 8, is also an* $\mathbf{SE}(3)$ *invariant probability path.*

The proposition could be proved following the Definition 4.3 and the Theorem 4.1. Combining the above terms, the final equivariant optimal transport based training objective is:

$$\mathcal{L}_{\text{CFM}}^{\text{EOT}}(\theta) = \mathbb{E}_{t,q(\mathbf{x}_0),p_1(\mathbf{x}_1)} \left\| v_\theta(\psi_t^{\text{EOT}}(\mathbf{x}_1),t) - (-\mathbf{x}_0 + (1-\sigma_{\min})\pi^*(\mathbf{R}^*\mathbf{x}_1)) \right\|^2 \quad (9)$$

A good property of the objective with EOT is that the training characteristics are invariant to translation and rotation of initial $\mathbf{x}_1$, and equivariant with respect to both sampled noise $\mathbf{x}_1$ and data point $\mathbf{x}_0$, which empirically contributes to more effective training.

**Solving Equivariant Optimal Transport**   We propose an iterative algorithm to obtain the equivariant optimal transport map. The algorithm first conducts the Hungarian algorithm [20] to align the atoms between the initial geometry from $p_1$ and the ground truth geometry from $p_0$; and then conducts the Kabsch algorithm [17] to solve the optimal rotation matrix based on the atom alignment. The proposed algorithm asymptotically converges to the optimal solution. Besides, the method holds a close relationship with the Iterative Closest Point (ICP) [5] algorithm, while our settings require the node alignment could only be one-one mapping. We leave the detailed description in Appendix C.

### 4.3   Information Aligned Hybrid Probability Path

In this section, we address the challenges posed by the multi-modality nature of 3D molecular data. Specifically, we focus on the distinct generation procedures required for various modalities, such as coordinates and atom types, within the flow-matching framework. It is crucial to recognize that altering atom types carries a different amount of chemical information compared to perturbing coordinates. To better understand this intuition, we provide the following corner case:

**Example 1:** $p_t(\mathbf{x}|\mathbf{x}_0) = p_0(\mathbf{x}|\mathbf{x}_0), \forall t < \epsilon_{\mathbf{x}}$ and $p_t(\mathbf{x}|\mathbf{x}_0) = p_1(\mathbf{x}|\mathbf{x}_0), \forall t \geq \epsilon_{\mathbf{x}}$.

We define the $p_t(\mathbf{h}|\mathbf{h}_0)$ similarly with a different parameter $\epsilon_{\mathbf{h}}$.Here we consider the corner case that $\epsilon_{\mathbf{x}} \to 0$ and $\epsilon_{\mathbf{h}} \to 1$, i.e. no noise for atom types from timestep 0 to timestep $\epsilon_{\mathbf{h}}$ and max noise level from $\epsilon_{\mathbf{h}}$ to timestep 1. (Reversely for $\epsilon_{\mathbf{x}}$ ) Under such a probability path, the model will be encouraged to determine and fix the node type at around $\epsilon_h$ step (very early step in the whole generation procedure), even if the coordinates are far from reasonable 3D structures. However, this particular case may not be optimal. The subsequent steps of updating the structure could alter the bonded connections between atoms, leading to a potential mismatch in the valency of the atoms with the early fixed atom types. Therefore, selecting a suitable inductive bias for determining the probability paths of different modalities is crucial for generating valid 3D molecules. In this paper, we utilize an information-theoretic inspired quantity as the measurement to identify probability paths for learning the flow matching model on 3D molecules.

**Definition 4.5.** For distribution $p_0$ on the joint space $\mathbf{g}$, and two corresponding conditional probability path $p_t(\mathbf{x}|\mathbf{g}_0)$ and $p_t(\mathbf{h}|\mathbf{g}_0)$, we denote the $I(\mathbf{x}_t, \mathbf{h}_t)$ as the mutual information for $\mathbf{x}_t$ with distribution $\int p_t(\mathbf{x}|\mathbf{g}_0)p_0(\mathbf{g}_0)\mathrm{d}\mathbf{g}_0$ and $\mathbf{h}_t$ with distribution $\int p_t(\mathbf{h}|\mathbf{g}_0)p_0(\mathbf{g}_0)\mathrm{d}\mathbf{g}_0$.

**Proposition 4.6.** *For the independent conditional probability path $p_t(\mathbf{g}|\mathbf{g}_0) = p_t(\mathbf{x}|\mathbf{x}_0)p_t(\mathbf{h}|\mathbf{h}_0)$, when the conditional probability path of $\mathbf{x}$ and $\mathbf{h}$ lies in OT path or VP path, if $I(\mathbf{x}_0, \mathbf{h}_0) > 0$, then $\forall t \in (0,1), I(\mathbf{x}_t, \mathbf{h}_t) > 0$ and $I(\mathbf{x}_{t_i}, \mathbf{h}_{t_i}) > I(\mathbf{x}_{t_j}, \mathbf{h}_{t_j}), \forall t_i < t_j$.*

We use the quantity $I_t(\mathbf{x}_t, \mathbf{h}_t)$ as the key property to distinguish different probability paths. Given the conditional probability path $p_t(\mathbf{x}|\mathbf{x}_0)$, it implies an information quantity change trajectory from $I(\mathbf{x}_1, \mathbf{h}_1) = 0$ to $I(\mathbf{x}_0, \mathbf{h}_0)$ following $I(\mathbf{x}_t, \mathbf{h}_0)$. Thus, one well-motivated probability path on $\mathbf{h}$ is to align the information quantity changes by setting $I(\mathbf{h}_t, \mathbf{h}_0) = I(\mathbf{x}_t, \mathbf{h}_0)$. Based on such intuition, we design our probability path on $\mathbf{h}$. We follow the data representation in [13]. This, for atom types, we represent it by one-hot encoding and charges are represented as integer variables. Empirically, the VP path involves a noise schedule function $\beta$ which could naturally adjust the information change by choosing different noise schedules, so we explore the probability path mainly on the VP path. For $I(\mathbf{h}_t, \mathbf{h}_0)$, we decompose it as $I(\mathbf{h}_t, \mathbf{h}_0) = H(\mathbf{h}_0) - H(\mathbf{h}_0|\mathbf{h}_t)$ where $H(\mathbf{h}_0)$ is constant and $H(\mathbf{h}_0|\mathbf{h}_t)$ is the conditional entropy towards $\mathbf{h}_0$ with $\mathbf{h}_t$ as the logits. Similarly, the difficulty of estimation $I(\mathbf{x}_t, \mathbf{h}_0)$ lies in $I(\mathbf{h}_0|\mathbf{x}_t)$. Following the difference of entropy estimator in [32], we build prediction model $p_\phi(\mathbf{h}_0|\mathbf{x}_t)$ to estimate $I(\mathbf{h}_0|\mathbf{x}_t)$ for selected time $t$. More details can be found in Appendix C. We demonstrate 20 time steps for $I(\mathbf{x}_t, \mathbf{h}_0)$, and $I(\mathbf{h}_t, \mathbf{h}_0)$ for vanilla VP path

Table 1: Results of atom stability, molecule stability, validity, and validity×uniqueness. A higher number indicates a better generation quality. The results marked with an asterisk were obtained from our own tests.

| # Metrics | QM9 | | | | DRUG | |
|---|---|---|---|---|---|---|
| | Atom Sta (%) | Mol Sta (%) | Valid (%) | Valid & Unique (%) | Atom Sta (%) | Valid (%) |
| Data | 99.0 | 95.2 | 97.7 | 97.7 | 86.5 | 99.9 |
| ENF | 85.0 | 4.9 | 40.2 | 39.4 | - | - |
| G-Schnet | 95.7 | 68.1 | 85.5 | 80.3 | - | - |
| GDM | 97.0 | 63.2 | - | - | 75.0 | 90.8 |
| GDM-AUG | 97.6 | 71.6 | 90.4 | 89.5 | 77.7 | 91.8 |
| EDM | 98.7 | 82.0 | 91.9 | 90.7 | 81.3 | 92.6 |
| EDM-Bridge | 98.8 | 84.6 | 92.0* | 90.7 | 82.4 | 92.8* |
| **EQUIFM** | **98.9** ± 0.1 | **88.3** ± 0.3 | **94.7** ± 0.4 | **93.5** ± 0.3 | **84.1** | **98.9** |

with the linear schedule ($VP_{linear}$) on $\beta$, VP path with cosine schedules($VP_{cos}$) [33] and polynomial schedules($VP_{poly}$) [13], and the OT path in Fig. 4.

We observe that the information quantity of $I(\mathbf{x}_t, \mathbf{h}_0)$ does not change uniformly, this is, it stays stable at the start and drops dramatically after some threshold. It is in line with the fact that when the coordinates $\mathbf{x}$ are away from the original positions to a certain extent, the paired distance between the bonded atoms could be out of the bond length range [6]. In this case, the point cloud $\mathbf{x}$ then loses the intrinsic chemical information. Reversely, the dynamics $I(\mathbf{x}_t, \mathbf{h}_0)$ also implies a generation procedure where the coordinates $\mathbf{x}$ transform first and the atom types $\mathbf{h}$ is then determined when $\mathbf{x}$ are relatively stable.

## 5 Experiments

In this section, we justify the advantages of the proposed EquiFM with comprehensive experiments. The experimental setup is introduced in Section 5.1. Then we report and analyze the evaluation results for the unconditional and conditional settings in Section 5.2 and 5.3. We provide detailed ablation studies in Section 5.4 to further gain insight into the effect of different probability paths. We leave more implementation details in Appendix B.4.

### 5.1 Experiment Setup

**Evaluation Task.** With the evaluation setting following prior works on 3D molecules generation [9, 30, 42, 13, 52], we conduct extensive experiments of EquiFM on three comprehensive tasks against several state-of-the-art approaches. *Molecular Modeling and Generation* assesses the capacity to learn the underlying molecular data distribution and generate chemically valid and structurally diverse molecules. *Conditional Molecule Generation* focuses on testing the ability to generate molecules with desired chemical properties. Following [13], we retrain a conditional version EquiFM on the molecular data with corresponding property labels.

**Datasets** We choose *QM9* dataset [39], which has been widely adopted in previous 3D molecule generation studies [9, 10], for the setting of unconditional and conditional molecule generation. We also test the EquiFM on the *GEOM-DRUG* (Geometric Ensemble Of Molecules) dataset for generating large molecular geometries. The data configurations directly follow previous work[3, 13].

### 5.2 Molecular Modeling and Generation

**Evaluation Metrics.** The model performance is evaluated by measuring the chemical feasibility of generated molecules, indicating whether the model can learn underlying chemical rules from data. Given molecular geometries, the bond types are first predicted (single, double, triple, or none) based on pair-wise atomic distances and atom types [13].

Next, we evaluate the quality of our predicted molecular graph by calculating both *atom stability* and *molecule stability* metrics. The atom stability metric measures the proportion of atoms that have a correct valency, while the molecule stability metric quantifies the percentage of generated molecules in which all atoms are stable. Additionally, we report *validity* and *uniqueness* metrics that indicate the

Table 2: Mean Absolute Error for molecular property prediction. A lower number indicates a better controllable generation result.

| Property Units | $\alpha$ Bohr$^3$ | $\Delta\varepsilon$ meV | $\varepsilon_{\mathrm{HOMO}}$ meV | $\varepsilon_{\mathrm{LUMO}}$ meV | $\mu$ D | $C_v$ $\frac{\mathrm{cal}}{\mathrm{mol}}$K |
|---|---|---|---|---|---|---|
| QM9* | 0.10 | 64 | 39 | 36 | 0.043 | 0.040 |
| Random* | 9.01 | 1470 | 645 | 1457 | 1.616 | 6.857 |
| $N_{\mathrm{atoms}}$ | 3.86 | 866 | 426 | 813 | 1.053 | 1.971 |
| EDM | 2.76 | 655 | 356 | 584 | 1.111 | 1.101 |
| **EQUIFM** | **2.41** | **591** | **337** | **530** | **1.106** | **1.033** |

Table 3: Ablation study, EquiFM models trained with different probability path, the effect of EOT is also evaluated.

| Method | Atom Stable (%) | Mol Stable (%) |
|---|---|---|
| EquiFM$_{\mathrm{EOT+VP_{Linear}}}$ | **98.9±0.1** | **88.3±0.3** |
| EquiFM$_{\mathrm{OT+VP_{Linear}}}$ | 98.7±0.1 | 84.9±0.4 |
| EquiFM$_{\mathrm{VP+VP_{Linear}}}$ | 98.4±0.1 | 81.6±0.3 |
| EquiFM$_{\mathrm{EOT+VP_{Cos}}}$ | 98.7±0.1 | 84.7±0.2 |
| EquiFM$_{\mathrm{EOT+VP_{Poly}}}$ | 98.7±0.1 | 83.4±0.5 |
| EquiFM$_{\mathrm{EOT+OT}}$ | 97.3±0.1 | 77.1±0.4 |

percentage of valid (determined by RDKIT) and unique molecules among all generated compounds. Furthermore, we also explore the sampling efficiency of different methods.

**Baselines.** The proposed method is compared with several competitive baselines. *G-Schnet* [9] is the previous equivariant generative model for molecules, based on autoregressive factorization. Equivariant Normalizing Flows (*ENF*) [42] is another continuous normalizing flow model while the objective is simulation-based. Equivariant Graph Diffusion Models (*EDM*) with its non-equivariant variant (*GDM*) [13] are recent progress on diffusion models for molecule generation. Most recently, [52] proposed an improved version of EDM (*EDM-Bridge*), which improves upon the performance of EDM by incorporating well-designed informative prior bridges. To yield a fair comparison, all the model-agnostic configurations are set as the same as described in Sec. 5.1.

**Results and Analysis.** We generate 10, 000 samples from each method to calculate the above metrics, and the results are reported in Table 1. As shown in the table, EquiFM outperforms competitive baseline methods on all metrics with an obvious margin. In the benchmarked 3D molecule generation task, the objective is to generate atom types and coordinates only. To evaluate stability, the bonds are subsequently added using a predefined module such as Open Babel following previous works. It is worth noting that this bond-adding process may introduce biases and errors, even when provided with accurate ground truth atom types and coordinates. As a result, the atom stability evaluated on ground truth may be less than 100%. Note the molecule stability is approximately the N-th power of the atom stability, N is the atom number in the molecule. Consequently, for large molecules in the GEOM-DRUG dataset, the molecule stability is estimated to be approximately 0%. Furthermore, as DRUG contains many more molecules with diverse compositions, we also observe that *unique* metric is almost 100% for all methods. Therefore, we omit the *molecule stability* and *unique* metrics for the DRUG dataset. Overall, the superior performance demonstrates EquiFM's higher capacity to model the molecular distribution and generate chemically realistic molecular geometries. We provide visualization of randomly generated molecules Appendix F and the efficiency study in Appendix E.

## 5.3 Controllable Molecule Generation

**Evaluation Metrics.** In this task, we aim to conduct controllable molecule generation with the given desired properties. This can be useful in realistic settings of material and drug design where we are interested in discovering molecules with specific property preferences. We test our conditional version of EquiFM on QM9 with 6 properties: polarizability $\alpha$, orbital energies $\varepsilon_{\mathrm{HOMO}}$, $\varepsilon_{\mathrm{LUMO}}$ and their gap $\Delta\varepsilon$, Dipole moment $\mu$, and heat capacity $C_v$. For evaluating the model's capacity to conduct property-conditioned generation, we follow the [42] to first split the QM9 training set into two halves with $50K$ samples in each. Then we train a property prediction network $\omega$ on the first half and train conditional models in the second half. Afterward, given a range of property values $s$, we conditionally draw samples from the generative models and then use $\omega$ to calculate their property values as $\hat{s}$. The *Mean Absolute Error (MAE)* between $s$ and $\hat{s}$ is reported to measure whether generated molecules are close to their conditioned property. We also test the MAE of directly running $\omega$ on the second half QM9, named *QM9* in Table 2, which measures the bias of $\omega$. A smaller gap with *QM9* numbers indicates a better property-conditioning performance.

**Baselines.** We incorporate existing EDM as our baseline model. In addition, we follow [13] to also list two baselines agnostic to ground-truth property $s$, named *Random* and $N_{atoms}$. *Random* means we simply do random shuffling of the property labels in the dataset and then evaluate $\omega$ on it. This operation removes any relation between molecule and property, which can be viewed as an upper bound of *MAE* metric. $N_{atoms}$ predicts the molecular properties by only using the number of atoms in the molecule. The improvement over *Random* can verify the method is able to incorporate conditional

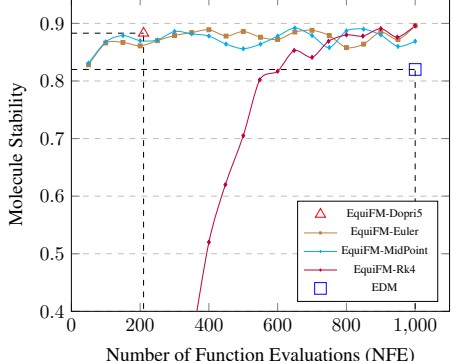
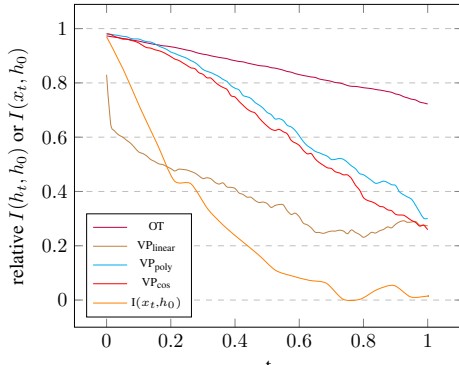

Figure 3: Resulting performance in molecule stability *w.r.t.* change of NFEs for each integration algorithm applied in the sampling process. EquiFM is better than our compared EDM model with every integration algorithm.

Figure 4: The x-axis is time, and the y-axis is the normalized mutual information estimation. It could be observed that $VP_{linear}$ path holds the closest tendency to that of $I(\mathbf{x}_t, \mathbf{h}_0)$, and OT path has the largest discrepancy.

property information into the generated molecules. And overcoming $N_{atoms}$ further indicates the model can incorporate conditioning into molecular structures beyond the number of atoms.

**Results and Analysis.** The visualizations of conditioned generation can be found in Appendix F. As shown in Table 2, for all the conditional generation tasks, our proposed EquiFM outperforms other comparable models with a margin. This further demonstrates the generalization ability of the proposed framework upon different tasks.

## 5.4   Ablations On the Impacts of Different Probability Paths

In this section, we aim to answer the following questions: 1) how is the impact of the different probability paths on the coordinate variable $\mathbf{x}$ and the categorical variable $\mathbf{h}$? 2) how does the equivariant optimal transport path boost the generation?

To answer these questions, we apply several different probability paths and compare them on the QM9 dataset, including the variance-preserving ($VP_{Linear}$) path(Eq. 5), vanilla optimal transport (OT) path(Eq. 4), and the equivariant optimal(Eq. 9) transport path(EOT) on the coordinate variable $\mathbf{x}$; And variance-preserving ($VP_{Linear}$) path(Eq. 5), vanilla optimal transport (OT) (Eq. 4), the variance-preserving path with polynomial decay ($VP_{Poly}$), variance preserving path with cosine schedule ($VP_{Cos}$). The result is illustrated in Tab. 3. We notice that OT-based paths on coordinates in general show superior performance than the others due to the stability and simplicity of the training objective. Furthermore, regularizing the path with the equivariant-based prior, the EOT path could further boost the performance by a large margin. To gain a more intuitive understanding, we further provide the generation dynamic comparison in Fig. 2(b). As shown, the generation procedure trained with vanilla OT path, though more stable than the EDM generation procedure, also exits some twisted phenomenon, *i.e.*, all atoms tend to first contract together and then expand; Such phenomenon disappears in the generation procedure of EOT path due to that the generation direction is well constrained. For the probability path on the categorical variable, we find the VP path, holds the superior performance due to the closest alignment with the information quantity changes. If there is a significant discrepancy in the information quantity dynamics, *e.g.*, OT path, it may result in a substantial decline in performance.

## 5.5   Sampling Efficiency

We also evaluate the sampling efficiency of our model, as shown in Fig. 3, the results of EquiFM with 4 different integrating algorithms converge to state-of-the-art results in much less NFE compared to baseline model EDM. Remarkably, the red triangle is the result of EquiFM with the Dopri5 integrating algorithm, it converges in approximately 210 NFE to achieve 0.883 model stability, while EDM takes 1000 NFE to achieve only 0.820. With simple non-adaptive step integration algorithms such as Eulers method and midpoint, the NFE required for convergence is much less than that of baseline models.

This indicates that our proposed model has learned a much better vector field, and takes a much shorter generation route during generation, this can be justified with visualization Fig. 2(a).

## 6 Conclusion and Future Work

We introduce the EquiFM, an innovative molecular geometry generative model that utilizes a simulation-free objective. While flow matching has demonstrated excellent properties in terms of stable training dynamics and efficient sampling in other domains, its application in geometric domains poses significant challenges due to the equivariant property and complex data modality. To address these challenges, we propose a hybrid probability path approach in EquiFM. This approach regularizes the probability path on coordinates and ensures that the information changes on each component of the joint path are appropriately matched. Consequently, EquiFM learns the underlying chemical constraints and produces high-quality samples. Through extensive experiments, we demonstrate that the EquiFM not only outperforms existing methods in modeling realistic molecules but also significantly improves sampling speed, achieving a speedup of **4.75**× compared to previous advancements. In future research, as a versatile framework, EquiFM can be extended to various 3D geometric generation applications, such as protein pocket-based generation and antibody design, among others.

## Acknowledgments

We thank the Reviewers for the detailed comments and Yanru Qu for the proofreading. This work is supported by National Key R&D Program of China (2022ZD0117502), Guoqiang Research Institute General Project, Vanke Special Fund for Public Health and Health Discipline Development, Tsinghua University (NO.20221080053), and Beijing Academy of Artificial Intelligence (BAAI), Tsinghua University (No. 2021GQG1012). Minkai Xu thanks the generous support of Sequoia Capital Stanford Graduate Fellowship.

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

# A  Sampling and Training Algorithm

We provide a detailed training and sampling pipeline in this section. The training algorithm with EOT path on the equivariant variable $\mathbf{x}$ and VP path on the invariant variable $\mathbf{h}$ as an example is demonstrated in Algorithm 1.

---

**Algorithm 1** Training Algorithm of EquiFM

---

1: **Input:** geometric data distribution $p_{\mathbf{g}}$, $\mathbf{g} = \langle \mathbf{x}, \mathbf{h} \rangle$
2: **Initial:** vector field network $v_\theta$, minimum constant variance $\sigma_{\min}$
3: **while** $\theta$ have not converged **do**
4:     $t \sim \mathbf{U}(0,1)$, $\epsilon \sim \mathcal{N}(\mathbf{0}, \boldsymbol{I})$, $g_0 \sim p_{\mathbf{g}}$
5:     Subtract center of gravity from $\epsilon_x$ in $\epsilon = [\epsilon_x, \epsilon_h]$
6:     Obtaining the Equivariant Optimal Transport plan $\pi^*, \mathbf{R}^*$ based on Algorithm 3
7:     $x_t = (\sigma_{\min} + (1 - \sigma_{\min})t)\pi^*(\mathbf{R}^* \epsilon_x) + (1 - t)x_0$               {Eq. 8}
8:     $h_t = \alpha_t h_0 + (1 - \alpha_t^2)\epsilon_h, \alpha_t = e^{-\frac{1}{2}T(t)}$                         {Eq. 5}
9:     $\mathcal{L}_{\text{EquiFM}} = ||v_\theta^x(\langle x_t, h_t \rangle, t) - (-x_0 + (1 - \sigma_{\min})\epsilon_x)||^2 + ||v_\theta^h(\langle x_t, h_t \rangle, t) - \frac{\alpha_t'}{1 - \alpha_t^2}(\alpha_t h_t - h_0)||^2$
10: **end while**
11: **return** $v_\theta$

---

The sampling algorithm can be found in Algorithm 2.

---

**Algorithm 2** Sampling Algorithm of EquiFM

---

1: **Input:** vector field model $v_\theta$
2: $g_1 \sim \mathcal{N}(\mathbf{0}, \boldsymbol{I})$
3: Subtract center of gravity from $x_1$ in $g_1 = [x_1, h_1]$
4: $g_0 = \text{ODESolve}(g_1, v_\theta, 1, 0)$
5: {During the ODE solving, always subtract the center of gravity from $v_\theta^x$}
6: sample $\hat{p}(x_0, h_0 | g_0)$
7: **return** $\langle x_0, h_0 \rangle$

---

Note the $\hat{p}(\cdot | g_0)$ stands for the procedure of transforming the continuous $h_0$ into the specific data modality. This is, for the categorical part, $\hat{p}(h | g_0) = \mathcal{C}(h | h_0)$ and for the integer part $p(h \mid g_0) = \int_{h - \frac{1}{2}}^{h + \frac{1}{2}} \mathcal{N}(u \mid h_0, \sigma_0) \, \mathrm{d}u$.

# B  Formal Proof of Theorems and Propositions

## B.1  Invariant Probability Path: Theorem 4.1

The key properties of the equivariant flow matching model here are the invariant density modeling. For simplicity, here we omit the invariant feature, *i.e.* $\mathbf{h}$, and focus on the variable $\mathbf{x}$. Here we demonstrated that with an invariant prior $p_1(\mathbf{x})$ and the equivariant vector field $v_\theta(\mathbf{x}, t)$, the marginal distribution implied by the vector field of each time step, $p_{\theta, t}(\mathbf{x})$ is also invariant.

*Proof.* We are given that $p_1(\mathbf{x})$ is invariant, and that $v_\theta(\mathbf{x}, t)$ is equivariant, *i.e.* for any rotation $\mathbf{R}$, $p_1(\mathbf{R}\mathbf{x}) = p_1(\mathbf{x})$, and $v_\theta(\mathbf{R}\mathbf{x}, t) = \mathbf{R}v_\theta(\mathbf{x}, t)$. Our target is to prove that $\forall t \in [0, 1]$, $\forall \mathbf{R}$, $p_{\theta, t}(\mathbf{R}\mathbf{x}) = p_{\theta, t}(\mathbf{x})$. Then specifically, the sampling distribution $p_0(\mathbf{x})$ is invariant.

First recall that the way we generate the distribution $p_{\theta, t}(\mathbf{x})$ is to exert a transformation $\psi_{\theta, t}$ to the prior $p_1(\mathbf{x})$. And the definition of the vector field is $v_\theta(\mathbf{x}, t) = \frac{\mathrm{d}}{\mathrm{d}t}\psi_{\theta, t}(\mathbf{x})$.

We can derive the equivariance of $\psi_{\theta, t}$ by the following equations:

$$\psi_1(\mathbf{x}) - \psi_{\theta,t}(\mathbf{x}) = \int_t^1 v_\theta(\mathbf{x}, t)\mathrm{d}t$$

$$\psi_1(\mathbf{R}\mathbf{x}) - \psi_{\theta,t}(\mathbf{R}\mathbf{x}) = \int_t^1 v_\theta(\mathbf{R}\mathbf{x}, t)\mathrm{d}t$$

$$= \int_t^1 \mathbf{R}v_\theta(\mathbf{x}, t)\mathrm{d}t$$

$$= \mathbf{R}(\psi_1(\mathbf{x}) - \psi_{\theta,t}(\mathbf{x}))$$

Note that $\psi_1(\mathbf{R}\mathbf{x}) = \mathbf{R}\psi_1(\mathbf{x})$ since $\psi_1(\mathbf{x}) = \mathbf{x}$, we have $\psi_{\theta,t}(\mathbf{R}\mathbf{x}) = \mathbf{R}\psi_{\theta,t}(\mathbf{x})$. That is to say, $\psi_{\theta,t}$ is an equivariant transformation. Thus its inverse $\psi_{\theta,t}^{-1}$ is also equivariant, since $\forall \mathbf{y} = \psi_{\theta,t}(\mathbf{x})$, we have $\mathbf{R}\mathbf{y} = \mathbf{R}\psi_{\theta,t}(\mathbf{x}) = \psi_{\theta,t}(\mathbf{R}\mathbf{x})$, so $\psi_{\theta,t}^{-1}(\mathbf{R}y) = \mathbf{R}\mathbf{x} = \mathbf{R}\psi_{\theta,t}^{-1}(\mathbf{y})$. Also, the Jacobian matrix $\frac{\partial \psi_{\theta,t}(\mathbf{x})}{\partial \mathbf{x}}$ is equivariant, *i.e.* $\frac{\partial \psi_{\theta,t}(\mathbf{u})}{\partial \mathbf{u}}\mid_{\mathbf{u}=\mathbf{R}\mathbf{x}} = \mathbf{R}\frac{\partial \psi_{\theta,t}(\mathbf{u})}{\partial \mathbf{u}}\mid_{\mathbf{u}=\mathbf{x}}$, which implies that $\det \frac{\partial \psi_{\theta,t}(\mathbf{u})}{\partial \mathbf{u}}\mid_{\mathbf{u}=\mathbf{R}\mathbf{x}} = \det \frac{\partial \psi_{\theta,t}(\mathbf{u})}{\partial \mathbf{u}}\mid_{\mathbf{u}=\mathbf{x}}$, since the $\det$ function keeps constant under any rotation.
According to the Change of Variable Theorem,

$$p_{\theta,t}(\mathbf{x}) = p_1(\psi_{\theta,t}^{-1}(\mathbf{x})) / \left| \det \frac{\partial \psi_{\theta,t}(\mathbf{u})}{\partial \mathbf{u}}\mid_{\mathbf{u}=\psi_{\theta,t}^{-1}(\mathbf{x})} \right|$$

$$p_{\theta,t}(\mathbf{R}\mathbf{x}) = p_1(\psi_{\theta,t}^{-1}(\mathbf{R}\mathbf{x})) / \left| \det \frac{\partial \psi_{\theta,t}(\mathbf{u})}{\partial \mathbf{u}}\mid_{\mathbf{u}=\psi_{\theta,t}^{-1}(\mathbf{R}\mathbf{x})} \right|$$

Applying the above conclusions together, we have

$$p_{\theta,t}(\mathbf{R}\mathbf{x}) = p_1(\mathbf{R}\psi_{\theta,t}^{-1}(\mathbf{x})) / \left| \det \frac{\partial \psi_{\theta,t}(\mathbf{u})}{\partial \mathbf{u}}\mid_{\mathbf{u}=\psi_{\theta,t}^{-1}(\mathbf{x})} \right|$$

$$= p_1(\psi_{\theta,t}^{-1}(\mathbf{x})) / \left| \det \frac{\partial \psi_{\theta,t}(\mathbf{u})}{\partial \mathbf{u}}\mid_{\mathbf{u}=\psi_{\theta,t}^{-1}(\mathbf{x})} \right| = p_{\theta,t}(\mathbf{x}).$$

$\square$

## B.2 Explanation of Proposition 4.2

Note for the initial prior distribution, $p_1(g|g_0)$, could be the standard distribution, *i.e.* $p_1(g|g_0) = \mathcal{N}(\mathbf{0}, \mathbf{I})$. In this case, $p_1(g|g_0) = p_1(x|x_0)p_1(h|h_0)$. And for the time step zero, if we assume the distribution is a Gaussian centralized on the $g_0$, *i.e.*, $p_0(g|g_0) = \mathcal{N}(g_0, \sigma_{\min}\mathbf{I})$. And in this case we also have that $p_0(g|g_0) = p_0(x|x_0)p_0(h|h_0)$.

**Example 2:** For the Gaussian probability path,

$$p_t(g \mid g_1) = \mathcal{N}\left(g \mid \mu_t(g_1), \sigma_t(g_1)^2 I\right) \tag{10}$$

where $\mu : [0,1] \times \mathbb{R}^d \to \mathbb{R}^d$ is the time-dependent mean of the Gaussian distribution, while $\sigma : [0,1] \times \mathbb{R} \to \mathbb{R}_{>0}$ describes a time-dependent scalar standard deviation (std).

The Gaussian probability path satisfies that $p_t(g|g_0) = p_t(x|x_0)p_t(h|h_0)$. To better clarify, we highlight the difference between the conditional probability path and the marginal probability path with the following Remark B.1.

*Remark* B.1. With the conditional probability path on $\mathbf{x}$ and $\mathbf{h}$ being independent of each other, the marginal distribution could be correlated, *i.e.* $p_t(g) \neq p_t(x)p_t(h)$

With this property, we could design different paths for modeling complex variables.

## B.3 Proof of Proposition 4.4

In the Proposition 4.4, we claim that the probability path under the EOT map is $\mathbf{SE}(3)$ invariant. The invariant property under translations is guaranteed by the training and sampling setting within Zero CoM space, and below we will focus on proving the path is invariant under rotations. The key observation is that $\mathbf{R}^*$ might be different for different $\mathbf{x}$.

Specifically, for any rotation $\mathbf{T}$ acting on the point cloud $\mathbf{x}$ with $N$ points, the new $\mathbf{R}^*$ corresponding with $\mathbf{Tx}$ (we denote it by $\mathbf{R}^*_{\text{rot}}$) exactly offsetting the impact of $\mathbf{T}$. Formally, let $\mathbf{x}_0$ denote the target point cloud, we calculate

$$(\pi^*, \mathbf{R}^*) = \operatorname*{argmin}_{(\pi, \mathbf{R})} \|\pi(\mathbf{Rx}^1, \mathbf{Rx}^2, \ldots, \mathbf{Rx}^N) - \mathbf{x}_0\|_2$$

$$(\pi^*_{\text{rot}}, \mathbf{R}^*_{\text{rot}}) = \operatorname*{argmin}_{(\pi, \mathbf{R})} \|\pi(\mathbf{RTx}^1, \mathbf{RTx}^2, \ldots, \mathbf{RTx}^N) - \mathbf{x}_0\|_2$$

We claim that $\pi^*_{\text{rot}} = \pi^*, \mathbf{R}^*_{\text{rot}} = \mathbf{R}^* \mathbf{T}^{-1}$, and a strict proof follows.

Note that if $\phi(\mathbf{R}) : \mathbb{R}^{3\times3} \to \mathbb{R}^{3\times3}$ is a reversible map, then for any scalar function $f(\mathbf{R})$, $\operatorname*{argmin}_{\mathbf{R}} f(\phi(\mathbf{R})) = \phi^{-1}(\operatorname*{argmin}_{\mathbf{R}} f(\mathbf{R}))$. Here let $\phi(\mathbf{R}) = \mathbf{RT}$, and $\phi^{-1}(\mathbf{R}) = \mathbf{RT}^{-1}$, we get

$$\begin{aligned}
(\pi^*_{\text{rot}}, \mathbf{R}^*_{\text{rot}}) &= \operatorname*{argmin}_{(\pi, \mathbf{R})} \|\pi(\mathbf{RTx}^1, \mathbf{RTx}^2, \ldots, \mathbf{RTx}^N) - \mathbf{x}_0\|_2 \\
&= \operatorname*{argmin}_{(\pi, \mathbf{R})} \|\pi(\phi(\mathbf{R})\mathbf{x}^1, \phi(\mathbf{R})\mathbf{x}^2, \ldots, \phi(\mathbf{R})\mathbf{x}^N) - \mathbf{x}_0\|_2 \\
&= \hat{\phi}^{-1}(\operatorname*{argmin}_{(\pi, \mathbf{R})} \|\pi(\mathbf{Rx}^1, \mathbf{Rx}^2, \ldots, \mathbf{Rx}^N) - \mathbf{x}_0\|_2) \\
&= \hat{\phi}^{-1}((\pi^*, \mathbf{R}^*)) = (\pi^*, \phi^{-1}(\mathbf{R}^*)) = (\pi^*, \mathbf{R}^* \mathbf{T}^{-1})
\end{aligned}$$

where $\hat{\phi}$ is an natural extension of $\phi$ defined as $\hat{\phi}((\pi, \mathbf{R})) = (\pi, \phi(\mathbf{R})), \forall(\pi, \mathbf{R})$.

Now we recheck the probability path $p_t$ in Eq. 8. Since $p_1$ is invariant under rotations, and the transformation $\psi_t^{\text{EOT}}$ satisfies

$$\begin{aligned}
\psi_t^{\text{EOT}}(\mathbf{x}) &= (\sigma_{\min} + (1 - \sigma_{\min})t)\pi^*(\mathbf{R}^*\mathbf{x}) + (1 - t)\mathbf{x}_0 \\
\psi_t^{\text{EOT}}(\mathbf{Tx}) &= (\sigma_{\min} + (1 - \sigma_{\min})t)\pi^*_{\text{rot}}(\mathbf{R}^*_{\text{rot}}\mathbf{Tx}) + (1 - t)\mathbf{x}_0 \\
&= (\sigma_{\min} + (1 - \sigma_{\min})t)\pi^*(\mathbf{R}^*\mathbf{x}) + (1 - t)\mathbf{x}_0
\end{aligned}$$

*i.e.* $\psi_t^{\text{EOT}}$ is invariant. So we conclude that $p_t = [\psi_t^{\text{EOT}}]_* p_1$ is also invariant under rotations.

## B.4 Explanation of Proposition 4.6

Here we provide the informal explanation of the Proposition 4.6. The proposition states that for OT path or VP path on $\mathbf{x}$ or $\mathbf{h}$, *i.e.* $p_t(x \mid x_0) = \mathcal{N}\left(x \mid (1 - t)x_0, (\sigma_{\min} + (1 - \sigma_{\min})t)^2 I\right)$ or $p_t(x \mid x_0) = \mathcal{N}\left(x \mid \alpha_t x_0, (1 - \alpha_t^2)I\right)$, the mutual information between the marginal variable $\mathbf{x}_t$ and $\mathbf{h}_t$ monotonically decays following the path from time step 0 to time step 1 where $I_0(\mathbf{x}_0, \mathbf{h}_0) > 0$. Note that the $I_1(\mathbf{x}_1, \mathbf{h}_1) = 0$, as $p_1(\mathbf{g}) = p_1(\mathbf{x})p_1(\mathbf{h})$. Recall the definition of signal-to-noise ratio (SNR) as:

$$\text{SNR} = \frac{\mu^2}{\sigma^2}$$

For OT-path, $\text{SNR} = \frac{(1-t)^2}{t^2}$; and for VP-path $\text{SNR} = \frac{\alpha_t^2}{1-\alpha_t^2}$. The key observation is that the SNR along the probability path of both $\mathbf{x}$ and $\mathbf{h}$ on either path will decay monotonically. Intuitively, with $t \to 1$, $\mathbf{x}_t$ has less information of $\mathbf{x}_0$ thus has less of $\mathbf{h}$.

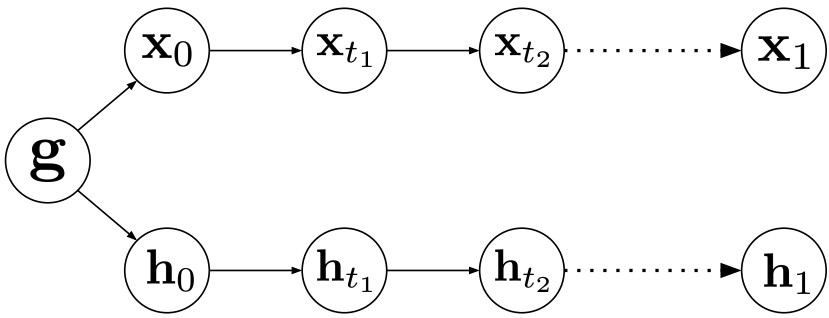

Figure 5: The dependency relationship on the hybrid path. Note that $0 < t_1 < t_2 < 1$, and the direction of arrows indicate the flow corresponding to the target vector field $u_t$ running backward (from $t = 0$ to 1, opposite of the sampling process) on a random molecule $\mathbf{g}$ (not in training data). This can be regarded as a process similar to that of a diffusion model, but $\mathbf{x}$ and $\mathbf{h}$ have conditionally independent paths.

### B.5 Proof of Proposition 4.6

*Proof.* Firstly, we will prove the dependency relationship between the time-dependent variables in the hybrid probability path as shown in Fig. 5.

To this end, we demonstrate the relationship between the $\mathbf{x}_0, \mathbf{h}_0, \mathbf{x}_{t_1}$. Recall the definition of conditional independent probability path as $p_t(\mathbf{x} \mid \langle \mathbf{x}_0, \mathbf{h}_0 \rangle) = p_t(\mathbf{x} \mid \mathbf{x}_0)$. Thus we have the distribution of random variable $\mathbf{x}_{t_1}$, $p_{t_1}(\mathbf{x}) = \int p_{t_1}(\mathbf{x} \mid \mathbf{x}_0) p(\mathbf{x}_0) \mathrm{d}\mathbf{x}_0$. And hence, there is $\mathbf{x}_{t_1} \perp \mathbf{h}_0 \mid \mathbf{x}_0$. Similarly, we could also derive $\mathbf{h}_{t_1} \perp \mathbf{x}_0 \mid \mathbf{h}_0$.

Next, for any time step $0 < t_1 < t_2 < 1$. We will then demonstrate the dependency relationship between $\mathbf{x}_0, \mathbf{x}_{t_1}, \mathbf{x}_{t_2}$. We denote the target vector field which generates the marginal probability path $p_t(\mathbf{x})$ as $u_t(\mathbf{x})$, then the distribution of random variable $\mathbf{x}_{t_2}$ could be then derived as $p_{t_2}(\mathbf{x}) = [\int_{t_1}^{t_2} u_t(\mathbf{x}) \mathrm{d}t]_* p_{t_1}(\mathbf{x}) = [\psi_{t_2}^u - \psi_{t_1}^u]_* p_{t_1}(\mathbf{x})$, where $\psi_t^u$ denotes the transformation corresponding with $u_t$. Then we have $\mathbf{x}_{t_2} \perp \mathbf{x}_0 \mid \mathbf{x}_{t_1}$. Similarly, $\mathbf{h}_{t_2} \perp \mathbf{h}_0 \mid \mathbf{h}_{t_1}$ could be also derived.

With the above two conclusions, we demonstrate the dependency relationship shown in Fig. 5 holds in the hybrid probability path. Therefore, with the dependency relationship we could directly obtain that $I(\mathbf{x}_{t_2}, \mathbf{h}_{t_2}) \leq I(\mathbf{x}_{t_1}, \mathbf{h}_{t_1})$. If $\alpha_t$ in VP path satisfies $\alpha_t > 0$ when $0 < t < 1$, then for any combination of such OT path and VP path, we always have $I(\mathbf{x}_t, \mathbf{h}_t) > 0$. $\square$

## C Implementation Details

### C.1 Solving EOT with a variant of iterative closest point (ICP) algorithm.

**Problem definition.** Given a point cloud $\mathbf{z} \in \mathbb{R}^{3 \times N}$ and its reference point cloud $\mathbf{y} \in \mathbb{R}^{3 \times N}$, note they are point cloud representations in Euclidean space. The objective is to find an optimal permutation matrix $\mathbf{\Pi}^* \in \mathbb{R}^{N \times N}$ and a rotation matrix $\mathbf{R}^* \in \mathbb{R}^{3 \times 3}$ that minimizes the following objective:

$$\mathbf{\Pi}^*, \mathbf{R}^* = \underset{\mathbf{\Pi}, \mathbf{R}}{\operatorname{argmin}} \| \mathbf{\Pi}(\mathbf{R}\mathbf{z})^\top - \mathbf{y}^\top \|_2 \tag{11}$$

We optimize the objective iteratively with a variant of the iterative closest point (ICP) algorithm, where it iteratively obtains $\mathbf{\Pi}$ and $\mathbf{R}$.

### C.2 Model Architectures and Training Configurations

We use the open-source software RDKIT [21] to preprocess molecules. For QM9 we take atom types (H, C, N, O, F) and integer-valued atom charges as atomic features, while for DRUG we only use atom types.

**Algorithm 3** A variant of iterative closest point (ICP) algorithm.

---
1: **Input:** a point cloud $\mathbf{z} \in \mathbb{R}^{3 \times N}$ and it's reference point cloud $\mathbf{y} \in \mathbb{R}^{3 \times N}$.
2: **while** $\tau$ has not converged **do**
3:     Obtain permutation matrix $\mathbf{\Pi} = \underset{\mathbf{\Pi}}{\mathrm{argmin}} \| \mathbf{\Pi}(\mathbf{Rz})^\top - \mathbf{y}^\top \|_2$ with Jonker-Volgenant algorithm [7]
4:     Obtain rotation matrix $\mathbf{R} = \underset{\mathbf{R}}{\mathrm{argmin}} \| \mathbf{R}(\mathbf{\Pi z})^\top - \mathbf{y}^\top \|_2$ with Kabsch algorithm [22]
5:     $\tau = \| \mathbf{\Pi}(\mathbf{Rz}^\top)^\top - \mathbf{y}^\top \|_2$
6: **end while**
7: **return** $\mathbf{\Pi}, \mathbf{R}$

---

The vector field network is implemented with EGNNs [43] by PyTorch [34] package. We set the dimension of latent invariant features $k$ to 1 for QM9 and 2 for DRUG, which extremely reduces the atomic feature dimension. For the training of vector field network $v_\theta$: on QM9, we train EGNNs with 9 layers and 256 hidden features with a batch size 64; and on DRUG, we train EGNNs with 4 layers and 256 hidden features, with batch size 64. The model uses SiLU activations. We train all the modules until convergence. For all the experiments, we choose the Adam optimizer [19] with a constant learning rate of $10^{-4}$ as our default training configuration. The training on QM9 takes approximately 2000 epochs, and on DRUG takes 20 epochs.

## D  Number of Evaluation (NFE) analysis

We further explore the behavior of adaptive integrators during sampling with Dopri15 as an example. In Fig. 7, we show the average NFE at different time intervals. We could observe that at time intervals near 0 the NFE is much larger than at other time intervals. The underlying reason lies in the vector field of $\mathbf{h}$ dramatically changes in these steps, which results in the frequent change of the atom type in the last period of sampling. This behavior could be due to the unsmoothness of the categorical manifold which could shed light on several future directions.

## E  Scalability

The proposed algorithm has a complexity of $O\left(n^2\right)$ for computing an OT map for a single molecule, where $n$ is the number of nodes. To understand the computational burden for different molecule sizes, we evaluated the average computing time for OT maps with varying node numbers. The Tab. 4 below shows the burden for three datasets/tasks. We also provide the curves of iteration and time needed to solve the EOT map in Fig. 6.

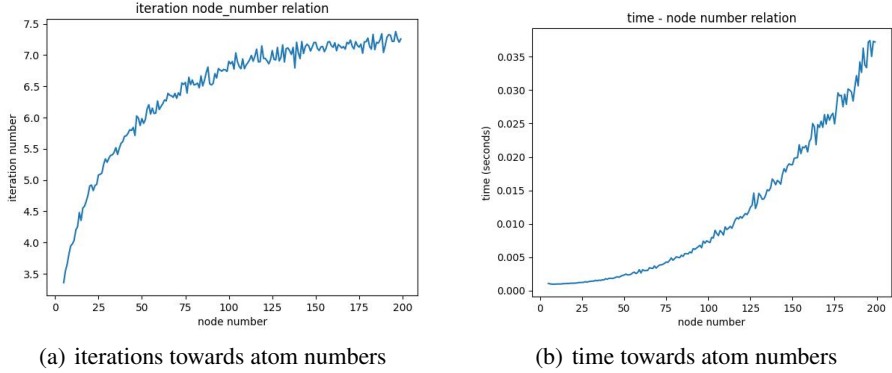

(a) iterations towards atom numbers        (b) time towards atom numbers

Figure 6: Time Cost for Solving EOT Maps

Even for the Antibody-CDR data, with an average of 150 atoms, the process time is only 18.84 ms, which is acceptable in practice. Additionally, we can optimize the process further by leveraging its

Table 4: Scalability to Different Molecule Size

|  | Average atom number | EOT mapping time per sample | EOT mapping iteration per sample |
|---|---|---|---|
| QM9 | 18 | 1.10ms | 4.67 |
| GEOM DRUG | 47 | 1.99ms | 5.89 |
| Antibody-CDR | 150 | 18.84ms | 7.14 |

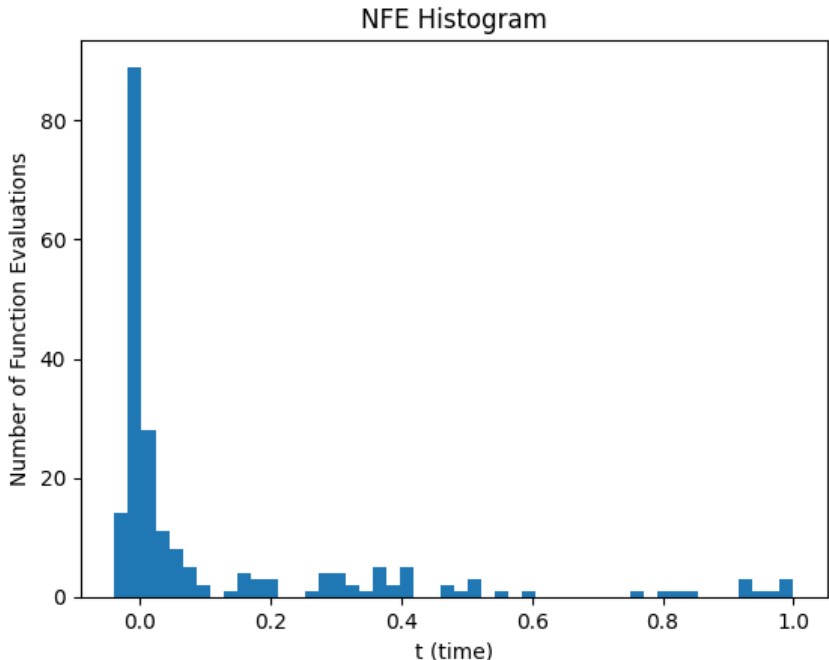

Figure 7: Number of Evaluation analysis of EquiFM generation process with Dopri5 integrator.

parallelizable nature. By Enabling prefetch and multiprocessing, we can minimize the computational overhead further and make it virtually inconsequential.

## F   More visualizations

This section presents additional visualizations of molecules generated by our EquiFM method. We include samples from two datasets, QM9 and DRUG, in Fig. 8 and Fig. **??**, respectively. All examples are randomly generated without cherry-picking, but the viewing direction may affect the visibility of some geometries.

As demonstrated in the figures, our model can generate realistic molecular geometries for small and large molecules alike. However, the model occasionally generates disconnected components, which is more common when trained on the large molecule DRUG dataset, as shown in the second molecule in Fig. **??**. This phenomenon is not unique to our model but is a common issue in non-autoregressive molecule generative models [54, 16]. Nevertheless, it is easily solvable by filtering out the smaller components.

We also present a qualitative assessment of controlled molecule generation by EquiFM in Fig. 9. We interpolate the conditioning parameter, polarizability, with different values of $\alpha$, while keeping the prior $g_1$ fixed. Polarizability measures the tendency of matter to acquire an electric dipole moment when subjected to an electric field. In general, less isometric molecular geometries tend to correspond to higher $\alpha$ values. This observation is consistent with our results in Fig. 9.

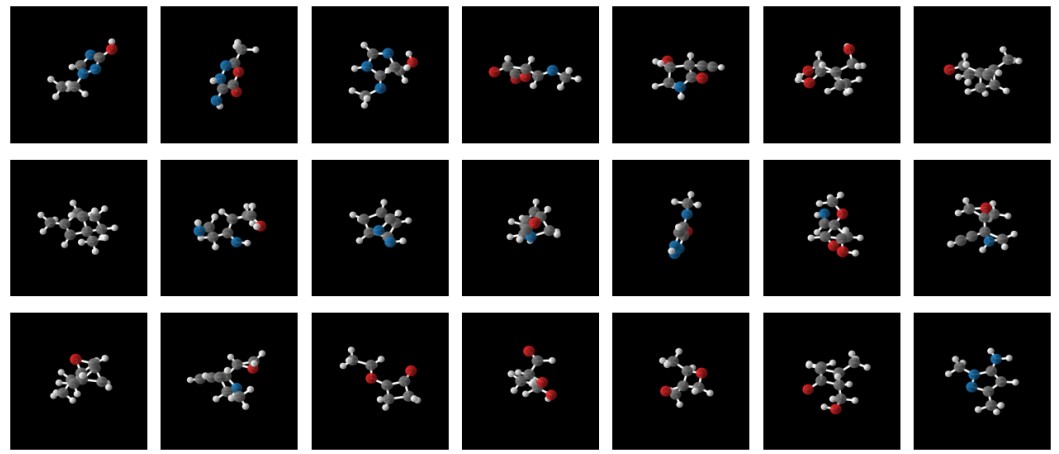

Figure 8: Molecules generated from EquiFM trained on QM9.

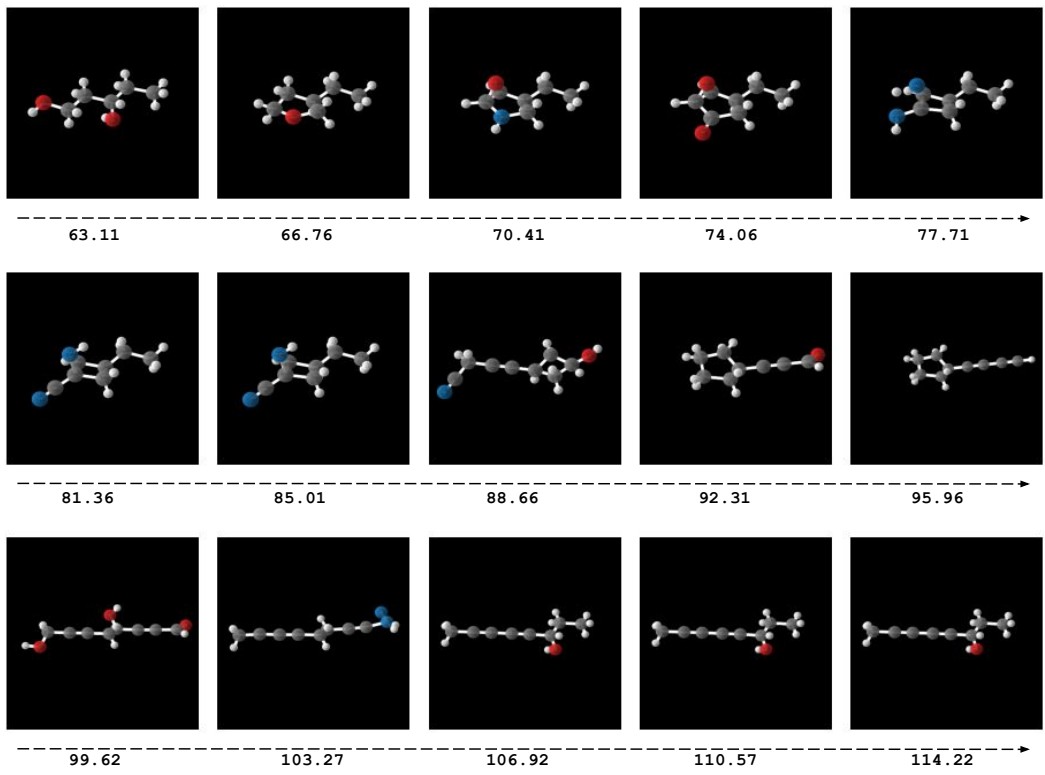

Figure 9: Molecules generated from conditioned version of EquiFM trained on QM9. We conduct controllable generation with interpolation among different polarizability $\alpha$ values with the same prior $g_1$. The given $\alpha$ values are provided at the bottom.

