# OpenReview forum: "Equivariant Flow Matching with Hybrid Probability Transport for 3D Molecule Generation"
_NeurIPS.cc/2023/Conference — NeurIPS 2023 poster_

### Official Review · Reviewer_KyMd · 2023-07-01

**Soundness:** 2 fair
**Presentation:** 1 poor
**Contribution:** 2 fair
**Rating:** 6
**Confidence:** 3

**Summary:**

The authors introduce a novel method named Equivariant Flow-Matching (EquiFM) for generating 3D molecules, aiming to enhance both categorical features (atom types) and continuous features (atom coordinates). The authors highlight the current limitations of diffusion models, particularly their instability and sampling inefficiencies. EquiFM improves upon these by using a flow-matching objective to stabilize the generative probability path of atom coordinates. Furthermore, it introduces a hybrid generative path to handle different modalities in the atomic feature space. This model utilizes an efficient ODE solver to enhance inference efficiency compared to existing SDE simulations. The authors report an improved performance with EquiFM, showing up to a 7% higher validity rate for large biomolecules and an average speed-up of 4.75x.

**Strengths:**

1. Equivariant Optimal Transport (EOT) is a novel approach to atom alignment, minimizing straight-line distance between paired atoms across all rotations and alignments.
2. EOT-based training objective is invariant to initial translations and rotations, increasing the model's robustness against variances in sampled noise and data points.
3. The iterative algorithm proposed for obtaining the EOT map is grounded in proven techniques (Hungarian and Kabsch algorithms), enhancing the solution's efficacy.
4. The unique approach of aligning information quantity changes in the probability paths for the variables ensures better modeling of the joint variable.

**Weaknesses:**


1. The application of the Hungarian and Kabsch algorithms for obtaining the EOT map may introduce computational overhead due to their iterative nature, affecting efficiency.
2. As the variables' probability paths are set independently, there might be cases when certain relationships or interactions between variables aren't accurately captured.
3. The document assumes prior knowledge of the topic and includes numerous technical terms and mathematical formulas, potentially making it inaccessible to a broader audience. For example, computational chemists and drug discovery scientists.
4. The use of a single rotation matrix for all possible rotations and alignments could potentially miss nuanced differences in complex data sets.
5. The model's complexity may lead to challenges in implementation, particularly in situations with less computational resources.

**Questions:**

1. What is the computational complexity of the EquiFM model, and how does it scale with increasing molecule size which limits the applicability of EquiFM for peptides and antibodies?
2. How might the performance of EquiFM change when applied to datasets with different distribution characteristics compared to QM9 and GEOM-DRUG?
3. What steps have been taken to ensure the chemical feasibility and synthetic accessibility of the generated molecules?
4. How was the model's hyperparameter configuration determined? Would a different configuration change the results significantly?
5. How would the model performance be affected if evaluated with other key metrics, like novelty, or drug-likeness?
6. Can the authors provide more detailed information on the hardware used for model training and inference, such as the specific GPU model, amount of RAM, and the number of parallel processes?
7. On what basis was the 4.75x speedup of the model calculated? Is this in comparison to a specific baseline model or an average across several? Could the authors elaborate on the exact methodology used for this calculation?

**Limitations:**

While the study introduces an innovative approach to the generation of 3D molecules provides a rigorous comparison with baseline models, there are several notable limitations that warrant further exploration.

1. A significant limitation of the current study is the relatively narrow scope of the references and baseline metrics utilized. Furthermore, the inclusion of more diverse baseline metrics would strengthen the validity of comparisons and aid in determining the true efficacy of the EquiFM model. For instance, discussing chemical similarity metrics between generated molecules and the training dataset could offer a more nuanced view of the model's capabilities.

2. The current evaluation metrics do not include measures such as synthetic accessibility, which assesses the ease with which a generated molecule can be physically synthesized in a laboratory.  Tools like RDKit offer a proxy metric to estimate synthetic accessibility (https://github.com/rdkit/rdkit/blob/master/Contrib/SA_Score/sascorer.py), and the lack of such consideration especially in relation to the DRUG dataset.

3. While the EquiFM model has shown promising results on the datasets used in this study, it remains unclear how it would perform on more diverse or complex molecular data. Expanding tests to include datasets like MOSES (https://github.com/molecularsets/moses) would allow a more comprehensive evaluation of the model's generalization capabilities.

In conclusion, while the EquiFM model has shown promise in generating 3D molecules, there is room for substantial expansion and improvement in future studies, particularly in the breadth of references and baseline metrics, and the inclusion of practical and diverse evaluation metrics.

---

> ### Author Rebuttal · Authors · 2023-08-10
>
> Thank the reviewer for the constructive and insightful comments. We address all your concerns about the computational complexity, the evaluation, and the details of the methods in the following paragraph, and any further discussions are welcome.
>
> ### Q1. Computational Complexity and Applicability for Larger Molecules
> Due to the page limit, the detailed response to the complexity issues could be referred to  Q1 of the response of Reviewer zq9q. And we provide extensive studies on the relationship between complexity and different molecule size. The full curve could be found in the uploaded pdf.
>
> ### Q2. Limitations on Capturing the Relationship with Independent Probability Paths
> According to the conditional flow matching theorem (Theorem.1 in [3]), one can utilize the conditional probability path $p(x_t,h_t|x_1, h_1)$ to learn the joint marginal flow $p(x_1，h_1)$. It should be emphasized that this statement is applicable regardless of any requirements on the distribution $p(x_1,h_1)$ or the correlation between $x$ and $h$. Additionally, there are no restrictions on the form of the conditional probability path, which allows for a valid and unbiased choice of learning any joint distribution with $p(x_t|x_1,h_1) = p(x_t|x_1)$. To provide further intuition, it is worth noting that the commonly used diffusion path for image generation also fits into the above formula when considering $x$ and $h$ as two pixels within the same image.
>
> ### Q3. The limitation of using a single rotation matrix
> As the reviewer mentioned, the single rotation matrix could lack the capacity to capture details when the point cloud holds more nodes and a complex structure. Fortunately, such limitation could be feasibly addressed by extending the proposed EOT map to a hierarchical version. For example, We could first compute the global EOT map with a rotation matrix to only align the center positions of fragments/scaffolds, And then calculate the local EOT map among the nodes inside each fragment.
>
> ### Q4. The performance change of EquiFM on distribution with different characteristics
> EquiFM is likely to excel in datasets with diverse conformation distributions due to the EOT map's ability to reduce learning space, minimizing optimization variance. However, modeling distributions where atom type and 3D structure are not strongly correlated, such as antibody variable regions, may pose challenges for the current version of EquiFM.
>
>
> ### Q5. How to ensure chemical feasibility and synthetic accessibility with EquiFM.
> Thus for a fair comparison with other fundamental models, we do not explicitly involve the components for chemical feasibility and synthetic accessibility in the current version. However, it could be very flexible for ensuring feasibility and synthetic accessibility with slight extra efforts based on EquiFM. For example, we could add an extra guided component on the vectors field like [2] to optimize the corresponding chemical feasibility and synthetic accessibility.
>
>
> ### Q6. Sensitivity to Hyperparameter settings
>
> We experimented with different settings for several optimization-related hyperparameters, e.g., learning rates and batch size, etc. And we found there is no significant impact of such hyperparameters.
>
>
> ### Q7. Detailed configuration of the hardware
> The configuration of our server is:
> CPU: Intel(R) Xeon(R) Platinum 8362 CPU @ 2.80GHz / 10 Core
> GPU: Nvidia-3090 GPU with 24 GB GPU memory / 1 GPUs
> Memory: 20G
>
>
> ### Q8. On what basis was the 4.75x speedup of the model calculated?
> Here the 4.75x speedup is compared to EDM[1], which is an advanced diffusion model for the task with promising performance. Note the EDM uses fixed diffusion steps, e.g., sampling every molecule will use 1000 forward calls of the EGNN in the paper.While for our model, the generation process is essentially solving the neural ODE of the learned vector field. With adaptive ODE solvers(dopri 15), such a process could be accelerated by jumping the repeated evaluation where the vector field changes are not significant. In this way, the adaptive ODE solver could gain acceleration. As the two models share the same EGNN architectures, we calculate the acceleration factor as the forward call number of EDM divided by that of EquiFM.
>
> ### Q9. More extensive evaluation metrics
> As suggested, we add the following extra evaluation metrics： Quantitative estimate of drug-likeness (QED),Retrosynthetic accessibility (RA)， Medicinal chemistry filter (MCF)， Synthetic accessibility score (SAS), Molecular weight(MW),LogP and novelty. Besides, we evaluate the Conformation Energy Distance to evaluate the quality of 3D conformations. The experimental results can be found in the following:
> | Methods | QED（$\uparrow$） | RA（$\uparrow$） | MCF （$\uparrow$） | SAS（$\downarrow$） | $$\Delta$$ MW（$\downarrow$） | $$\Delta$$ LogP （$\downarrow$） | Conformation Energy distance（$\downarrow$） | Novelty $\uparrow$） |
> | --- | --- | --- | --- | --- | --- | --- | --- | --- |
> | EDM | 0.608 | 0.441 | 0.621 | 4.054 | 0.566 | 23.71 | 0.2180 | 0.791 |
> | EquiFM | 0.627 | 0.519 | 0.693 | 3.893 | 0.478 | 19.54 | 0.2081 | 0.834 |
>
> Here $\Delta$ stands for the difference compared to ground truth distribution. And here to get valid molecules on GEOM-DRUGs, for both models, we sample Hydrogen coordinates with the help of RDkit following previous work [1].
>
>
> ### Q10. Extended the experimental results on other datasets
>  After carefully checking the data and introduction in MOSEs, we find that the MOSEs mainly contain 2D information, e.g., SMILES and atom-bond graphs while the 3D conformation data is missings. Therefore, it would be not suitable to evaluate the proposed EquiFM which is a 3D molecule generative model.
>
> [1]. Hoogeboom, Emiel, et al, Equivariant diffusion for molecule generation in 3d.ICML 2022.
> [2]. Bao et al. Equivariant Energy-Guided SDE for Inverse Molecular Design. ICLR 2023
> [3] Lipmon et al, Flow Matching for Generative Modeling. ICLR 2023

---

> > ### Comment · Reviewer_KyMd · 2023-08-16
> >
> > Thank you for providing a comprehensive and detailed rebuttal to the review. The extensive explanations and additional experimental results presented in the rebuttal have effectively addressed the issues raised in the initial review. Based on the responses and the additional information provided, I am convinced of the technical soundness and contribution of the work therefore I am willing to raise the score to 6

---

> > > ### Author Response · Authors · 2023-08-17
> > > **Thanks for your feedback!**
> > >
> > > Thank you very much for recognizing our work and providing valuable feedback!  If there is any additional information that you might need, please don't hesitate to inform us.

---

### Official Review · Reviewer_EmYL · 2023-07-02

**Soundness:** 3 good
**Presentation:** 1 poor
**Contribution:** 3 good
**Rating:** 7
**Confidence:** 3

**Summary:**

This paper addresses the molecular generation problem.
The authors propose a conditional flow-matching-based method that employs different generative paths for coordinates and node-wise features.
In addition, EGNN is used for the SE(3) invariant vector field, and an ICP-like algorithm is used for the equivariant optimal transport map.
The proposed method outperforms the conventional method on the molecule generation benchmark tasks.

**Strengths:**

* The hybrid probability path is an attractive solution for the generative probability path of different modalities.
* The equivariant OT map is a reasonable approach for 3D coordinate variables.
* The proposed method can generate molecules not only with better quality but also faster.
* The proposed method also outperforms a conventional method in conditional generation leading to many industrial applications.

**Weaknesses:**

* Since the proposed method is not clearly stated, the ablation study is not easy to understand, such as (E)OT+VP_{xxx} in Table 3.
* The model architecture is poorly explained, so the generation of discrete variables, such as atomic types, is difficult to understand.
* Some TeX references in the main text need to be corrected, such as Tab. 5.2 --> Table 3 in Section 5.4,  Fig. 5.4 --> Fig. 3 in Section 5.5, and Appendix C --> B.4 in Section 4.3.

**Questions:**

* Do you have any original definitions in Section 3? If you do not mean to argue your originality in the section, it is better to clearly state that the section is a review of [26].
* What is the definition of $\sigma_{min}$?
* Some contents seem to be missing. For example, the detail of Section 4.3 is not included in Appendix C, although there is a reference in the main text.
* Does EOT+VP_{linear} in Table 3 mean EOT path on x and VP_{linear} on h?  I cannot find the definition of + in (E)OT+VP_{xxx}.
* Please explain how you generated atomic types in your model. Did EGNN output a one-hot vector representing atomic types?

**Limitations:**

The authors did not address the limitations.

---

> ### Author Rebuttal · Authors · 2023-08-10
>
> Thank you very much for the detailed and insightful comments! The responses to your concerns are listed below:
>
> ### Q1. The ablation study is not easy to understand, such as (E)OT+VP_{xxx} in Table 3
> As mentioned by the reviewer, in Table 3, the term "$EOT+VP_{linear}$" does refer to the method where the EOT path is applied to x and $VP_{linear}$ is applied to h. Thank you for bringing this to our attention. We will ensure that the corresponding sections are carefully proofread in order to eliminate any confusion in the next version.
>
> ### Q2. The model architecture is poorly explained.  The generation of discrete variables, such as atomic types, is difficult to understand
>
> We apologize for the confusion we may have caused. The model architecture/parameterization closely adheres to the EDM [1] paper, in order to ensure a fair comparison of the impact of the new training objective. During the generation of atom types, the EGNN produces continuous vectors at each timestep. The only additional step involved in generating discrete variables is the application of a quantized operation, such as $argmax$, to transform them into discrete vectors.
>
> ### Q3.  Missing TEX references and contents, e.g. Details of section 4.3.
> Thank you for bringing this to our attention. We acknowledge that the missing details of Section 4.3 can be found in Appendix B.4, as correctly pointed out by the reviewer. We will thoroughly review and address the reference issues mentioned, such as replacing Tab. 5.2 with Table 3 in Section 5.4, updating Fig. 5.4 to Fig. 3 in Section 5.5, and correcting the reference to Appendix C to be B.4 in Section 4.3. These corrections will be diligently made in the updated version.
>
> ### Q4. it is better to clearly state that the section is a review of [26]
> Thank you for your suggestion, which we greatly appreciate. The content related to flow-matching in Section 3 serves the purpose of providing essential background information and introducing relevant notations. In response to the reviewer's suggestion, we will explicitly clarify in the section that this content is intended to provide foundational knowledge rather than claim originality. By doing so, we hope to eliminate any potential misunderstanding regarding our contributions. Thank you for pointing this out, and we will make the necessary clarification in the revised version.
>
> ### Q5. What is the definition of $\sigma_{min}$
> Please note that the introduction of $\sigma_{min}$ serves the purpose of approximating each data point, specifically a delta distribution, by a Gaussian distribution with a narrow peak centered at the data point and a very small variance ($\sigma_{min}$). This approximation is employed to prevent any corner cases and to facilitate a simpler mathematical representation of the probability path, as demonstrated in [1]. In response to the suggestion, we will enhance clarity by including this explanation in the upcoming version.
>
> [1] Lipmon et al, Flow Matching for Generative Modeling. ICLR 2023

---

> > ### Comment · Reviewer_EmYL · 2023-08-19
> > **Thank you**
> >
> > I appreciate the response from the authors.  The answers to my questions were satisfying, and I look forward to seeing the corrections of the pointed-out errors and additional explanations.
> >
> > I have also read the opinions of the other reviewers. The weaknesses of this paper primarily lie in the insufficient explanation.
> > However, given its strong technical contribution, major revisions are not required.
> > I believe this paper should be accepted, and I have raised my evaluation.

---

> > > ### Author Response · Authors · 2023-08-19
> > > **Thank you very much for your feedback!**
> > >
> > > Thank you sincerely for your valuable feedback and recognition of our work! We want to assure you that we will address the insufficient explanation and fix the pointed-out errors as suggested in the next version. If there is any further information you may need, please feel free to let us know!

---

### Official Review · Reviewer_tN9X · 2023-07-06

**Soundness:** 2 fair
**Presentation:** 1 poor
**Contribution:** 2 fair
**Rating:** 3
**Confidence:** 4

**Summary:**

The contribution is a new 3D generative model for molecules. The model is trained using a novel flow-matching objective. The flow-matching objective for coordinates of atoms is novel in that coordinates of 'source' atoms are permuted and rotated to align with 'target' atoms. There is also some exploration of how to set probability paths for different parts of the molecule description, namely the atom types, charges, and coordinates.

**Strengths:**

Flow-matching represents a very promising technique to improve on generative models like Hoogeboom et al.'s EDM, in terms of both training and sampling speed.  The idea of permuting and rotating source atom coordinates to straighten the flow that is matched is interesting, as is the question of how to set the relative corruption rates for different parts of a compound data type (coordinates, atom types, and charges). Figure 2 is beautiful.

**Weaknesses:**

The paper is not clearly written, and does not seem to have been proof-read (e.g., repetition in lines 26-29).

Section 4.3 describes what one might try to achieve when choosing a probability path for $h$ but does not say how to do it. Is the answer that you calculate and plot the lines in Figure 4 and then pick the path whose line is visually closest to the $I(x_t, h_0)$ line?

The ablation studies are not thorough. In particular, permuting and rotating atom coordinates as in definition 4.3 is a key novel feature of the proposed model, and the authors should show quantitatively what effect it has on the quality of generated samples.  Line 331 refers to 'Tab 5.2' but I cannot find the table this refers to.


**Questions:**

Lines 288-289: it looks like some words are missing ('the achieves' does not make grammatical sense) but it seems the authors are claiming that in GEOM-DRUGS, almost every molecule has one or more atoms with incorrect valency. Is that really true? I thought that GEOM-DRUGS was a collection of drug-like molecules in realistic conformations.

In figure 3, it's surprising that RK4 with 4x the number of evaluations usually does worse than Euler. Why is RK4 so bad?

In section 4.2 please could the authors spell out what is claimed to be equivariant with respect to what?

Equation (8): this $\psi_t$ will have discontinuities with respect to $x$ at the values of $x$ where $\pi$ changes. Is this a problem?

How are atom types and charges represented in $h$?  Are they one-hot encoded as in Hoogeboom et al's EDM?



**Limitations:**

Among the limitations of previous models that the authors say they address is the inability to generate large molecules. However, the paper does not show any good generated molecules with more than 9 heavy atoms.

---

> ### Author Rebuttal · Authors · 2023-08-10
>
> We thank the reviewer for the valuable feedback and appreciate the effort in reviewing our work. We will address your concerns on the ablation studies, the evaluations, and the method details in the following paragraph, and any further comments are welcome!
>
> ### Q1. Presentations and typos in Line26-Line29:
> Thanks for pointing it out. We will carefully proofread the draft as suggested in the updated version.
>
> ### Q2. Section 4.3 describes what one might try to achieve when choosing a probability path for $h$ but does not say how to do it.
>
> To clarify, the key ingredient of this section is proposing the use of different probability paths for different modalities. We introduce an inductive bias by aligning the mutual information term $I(h_t,h_0)$ with $I(x_t,h_0)$. This is based on the intuition that molecular/chemical information emerges when the coordinates (x_t) are within a certain range, such as bond distances, to the ground truth. We believe the probability path of atom types $h_t$ should also reflect this emergence of information. Thus, we use mutual information as a measurable quantity to describe this tendency. Matching the tendencies of the two curves is sufficient to achieve appealing results, considering the uncertainty/bias in approximating mutual information $I(x_t,h_0).
>
> Following the reviewer's suggestion, we will introduce some quantity metrics to make the approach more rigorous in the next version. We have calculated several quantities to measure the tendencies of different probability trajectories, and these will be incorporated into the updated version.
>
>
> ### Q3. In section 4.2 please could the authors spell out what is claimed to be equivariant with respect to what?
> Apologies for the confusion caused. In section 4.2, the equivariant optimal transport is defined as a point mapping between two point clouds that is optimal for all E(3) equivariant operations on either point cloud. We understand that this definition can be misleading in terms of the method name (Equivariant Flow Matching), which actually refers to the fact that the modeled vector field is equivariant towards E(3) operations on the input. To avoid any misunderstanding, we will provide additional clarifications to make this distinction clear in the next version.
>
> ### Q4. RK4 with 4x the number of evaluations usually does worse than Euler. Why is RK4 so bad?
> For Euler, one step of integration requires only one NFE, while 4 NFEs are needed for one step of RK4 as it's a fourth-order Runge-Kutta method. For sufficient step time, the RK4 could be better than Euler as in Figure 3.  The performance drop when the step is insufficient could be due to the discrete property of $h$
>
> ### Q5. $\psi_t$ in Eq.8 will have discontinuities with respect to x where $\pi$ changes. Is it a problem？
> This is indeed a great question.
> Firstly, the $\psi_t$ in Eq. 8 is a valid path for training continuous normalizing flows. This is due to the fact that $\psi_t$ is time-dependent diffeomorphic [1], i.e., there exists a continuous vector field $v_t$ satisfies that $\frac{d}{d t} \psi_t(x)=v_t\left(\psi_t(x)\right)$. Here $v_t = \pi^*\left(\mathbf{R}^* \mathbf{x}_1\right)- \mathbf{x}_0$.
> Meanwhile, as the reviewer mentioned, this $\psi_t$ could have discontinuities with respect to x. This could hurt the generalization/robustness during sampling though the objective is unbiased. And we believe that regularizing the discontinuities could be a reasonable future direction to explore for better performance.
>
> [1] Lipmon et al, Flow Matching for Generative Modeling. ICLR 2023
>
> ### Q6. How are atom types and charges represented in $h$? Are they one-hot encoded as in EDM?
> Yes, we follow the data representation as in EDM. This, for atom types, we represent it by one-hot encoding and charges are represented as integer variables.
>
> On the Evaluation and Extra Ablation Studies
>
> ### Q7. The ablation studies are not thorough. In particular, the quantitative results of EOT maps.
> Thanks for pointing it out, as you mentioned, the 'Tab 5.2' Line 331 is actually Table 3. In Table 3, we actually conduct ablation studies on the effect of EOT maps by fixing the probability path on $h$ as the $VP_{\text{Linear}}$ path, while enumerating the probability path on atom coordinates as $EOT$, $OT$, $VP$. As we could find, the $EOT$ map could consistently bring performance improvements upon that without the $EOT$ map, e.g. vanilla $OT$ path or $VP$ path. To make the ablation studies more comprehensive, we add extra ablations in the following:
>
>
> | Metrics | $EOT+VP_{cos}$ | $EOT + VP_{sin}$ | $OT+VP_{cos}$ | $OT+VP_{sin}$ |
> | --- | --- | --- | --- | --- |
> | Atom Stability | 98.7 | 98.7 | 97.9 | 97.7 |
> | Mol Stability | 84.7 | 83.4 | 80.1 | 79.8 |
>
>
> We will update the ablation studies in the updated version.
>
> ### Q8. It is claimed that in GEOM-DRUGS, almost every molecule has one or more atoms with incorrect valency.
> As the reviewer mentioned, the GEOM-DRUGS dataset comprises realistic conformations of drug-like molecules. In this context, it is important to clarify that we are not suggesting any incorrect valency in the real molecules within GEOM-DRUGS. The objective of the benchmarked 3D molecule generation task is to generate atom types and coordinates, followed by the addition of bonds using a predefined module [1,2].
>
> However, the process of adding bonds may introduce some bias. Consequently, using the same bond-adding process will result in ground truth data with atom stability lower than 100%. It's worth noting that molecule stability is approximately calculated as the N-th power of atom stability, where N represents the number of atoms in the molecule. Thus, for large molecules in the GEOM-DRUGS dataset, molecule stability is estimated to be close to 0%.
>
> [1]. Hoogeboom, Emiel, et al, Equivariant diffusion for molecule generation in 3d. ICML 2022.
>
> [2]. Wu, et al. Diffusion-based Molecule Generation with Informative Prior Bridges. NeurIPS 2022.

---

> > ### Comment · Reviewer_tN9X · 2023-08-16
> >
> > Thank you for the response. I like the main idea of the paper but I feel that the presentation is too poor to recommend acceptance.

---

> > > ### Author Response · Authors · 2023-08-18
> > > **Thanks a lot for your feedback!**
> > >
> > > We thank the reviewer for the response and valuable feedback. We assure you that the paper will be carefully proofread and revised as suggested. To this end, we provide the main revisions in the following:
> > > - Line26-Line29 in the Introduction _"However, ......, the empirical evaluation metrics such as validity, stability, and molecule size."_ would be updated as:
> > >   "However, despite great potential, the performance is indeed limited considering several important empirical evaluation metrics such as validity, stability, and molecule size, due to the insufficient capacity of the  underlying generative models."
> > > - Before Line 188, _"The equivariant optimal transport .....  rotations and alignment."_ We will add the following sentences:
> > > "With $\mathbf{y}$ and $\mathbf{z}$ lie in the zero of mass space,  the mappings $\pi^{*}$ in Eq. 7 are optimal towards any  E(3) equivariant operations on either side of the point clouds.  Therefore, the mappings are referred to as equivariant optimal transport(EOT)."
> > > - Section 3 will add statements to clarify the contribution before Line 92:
> > >   "In this section, we provide an overview of the general flow matching method to introduce the necessary notations and concepts based on [1]."
> > > - Line 119-Line 120, _"With the prior distribution_ ........ $\mathcal{N}\left(x\mid x_0,\sigma_{\min}^2 I\right)$", will be revised as:
> > >       With the prior distribution $p_1$ defined as a standard Gaussian distribution,  the empirical data distribution  $p_0(x\mid x_0)$ is approximated with a peaked Gaussian centered in  $x_0 $ with a small variance $\sigma_\text{min}$ as $\mathcal{N}\left(x\mid x_0,\sigma_{\min}^2 I\right)$.
> > > - Section 4.3 will be revised in line with the suggested changes
> > >   - Line 209-Line 213, _"With the conditional, ......., the following examples"_,  will be updated as: "In this section, we address the challenges posed by the multi-modality nature of 3D molecular data. Specifically, we focus on the distinct generation procedures required for various modalities, such as coordinates and atom types, within the flow-matching framework. It is crucial to recognize that altering atom types carries a different amount of chemical information compared to perturbing coordinates.  To better understand this intuition, we provide the following corner case:"
> > >   - Line 217-Line 221, _"Now consider the case ........ quantity"_ will be revised as:
> > >       "Here we consider the corner case that $\epsilon_\textbf{x} \to 0$ and $\epsilon_\textbf{h} \to 1$, i.e. no noise for atom types from timestep 0 to timestep $\epsilon_\textbf{h}$ and max noise level from $\epsilon_\textbf{h}$ to timestep 1. (Reversely for $epsilon_\textbf{x}$)  Under such a probability path, the model will be encouraged to determine and fix the node type at around $\epsilon_h$ step (very early step in the whole generation procedure), even if the coordinates are far from reasonable 3D structures.  However, this particular case may not be optimal. The subsequent steps of updating the structure could alter the bonded connections between atoms, leading to a potential mismatch in the valency of the atoms with the early fixed atom types.
> > >        Therefore, selecting a suitable inductive bias for determining the probability paths of different modalities is crucial for generating valid 3D molecules. In this paper, we utilize an information-theoretic inspired quantity as the measurement to identify probability paths for learning the flow matching model on 3D molecules."
> > >   - Line 232,  after _"we design our probability path on $h$"_:
> > >       "We follow the data representation in [2]. This, for atom types, we represent it by one-hot encoding and charges are represented as integer variables."
> > > - Line 288-Line 291 in Section 5.2: _"It is worth noticing that, ....., and distances."_ Would be updated as:
> > >        "In the benchmarked 3D molecule generation task, the objective is to generate atom types and coordinates only.  To evaluate stability, the bonds are subsequently added using a predefined module such as Open Babel following previous works.  It is worth noting that this bond-adding process may introduce biases and errors, even when provided with accurate ground truth atom types and coordinates. As a result, the atom stability evaluated on ground truth may be less than 100%.  Note the molecule stability is approximately the N-th power of the atom stability, N is the atom number in the molecule. Consequently, for large molecules in the GEOM-DRUG dataset, the molecule stability is estimated to be approximately 0%."
> > > - The TEX reference errors are fixed:
> > >   - Tab 5.2 in Line 331 is corrected as Tab 3.
> > >   - Fig. 5.4 in Line 343 is corrected as Fig. 3.
> > >   - Appendix C in Line 238 is corrected as  Appendix B.4
> > >
> > > [1].  Lipmon et al, Flow Matching for Generative Modeling. ICLR 2023
> > >
> > > [2] Hoogeboom, Emiel, et al, Equivariant diffusion for molecule generation in 3d. ICML 2022.
> > >
> > > Please feel free to let us know if you need any additional information!

---

### Official Review · Reviewer_zq9q · 2023-07-06

**Soundness:** 3 good
**Presentation:** 3 good
**Contribution:** 3 good
**Rating:** 7
**Confidence:** 4

**Summary:**

The paper applies the Flow Matching framework for 3D molecule generation achieving state of the art performance on common benchmarks. The authors introduce several innovations to the FM framework to adjust it for the equivariant data setting with different data modalities within a single sample (e.g., 3D coordinates, atom types, charge).  The two main modifications are (i) applying OT alignment between coordinates to shorten learned paths. (Similar to [1,2]), (ii) using different probability paths for different modalities in the data.
The proposed method improves both sampling speed and performance and achieves SOTA in both unconditional and conditional generation.

**Strengths:**

- The paper is well written. The problem setting, the motivation and the proposed method are introduced clearly.
- The paper showcases an application of FM to equivariant data domain.
- Achieve SOTA performance on common benchmarks.
- Experimental section is thorough and presents ablations justifying the algorithmic choices made and demonstrating the benefits in the flexibility of the FM framework.

**Weaknesses:**

- **Scalability** - in the paper the authors demonstrated the OT alignment on small molecules, how would the method scale to larger sets? How computationally prohibitive it is to compute the OT maps?
- **$S_n$ invariance** - A proof that the probability path implied by the EOT map is also $S_n$-invariant is missing from proposition 4.4. Meaning that for permutation of the atoms in the molecule the same probability will be returned.


**Questions:**

- Regarding the information alignment -  have the authors tried to scale one of the modalities to have the same variance as the other? Can the authors provide some more motivation for using the mutual information?


**Missing related work**
I ask the authors to a add discussion on the following missing related works -

- Two closely related prior/concurrent works that applied OT alignment to improve sampling speed:

  - [[1]](https://arxiv.org/abs/2304.14772) Multisample Flow Matching: Straightening Flows with Minibatch Couplings, Pooladian et. al. (ICML 2023)

  - [[2]](https://arxiv.org/abs/2302.00482) Conditional Flow Matching: Simulation-Free Dynamic Optimal Transport, Tong et. al. (Preprint)

- Another concurrent work pursuing the same applications:

  - [[3]](https://arxiv.org/abs/2305.01140) Geometric Latent Diffusion Models for 3D Molecule Generation, Xu et. al. (ICML 2023)

** Note: These works do not diminish the contribution of this work. It is however necessary to mention and discuss the differences for the completeness of the paper.

**Limitations:**

limitations are not discussed.

---

> ### Author Rebuttal · Authors · 2023-08-10
>
> We thank the reviewer for the insightful comments and the recognization of our work. And we address your concerns in the following:
> ### Q1. How would OT maps scales to larger sets and how computationally prohibitive is it?
>
> I first want to clarify that OT maps are not needed during inference. And it is true that OT maps add extra overhead during training. However, this does not limit scalability for larger sets.
>
> The proposed algorithm has a complexity of $O(n^2)$ for computing OT map for a single molecule, where $n$ is the number of nodes. To understand the computational burden for different molecule sizes, we evaluated the average computing time for OT maps with varying node numbers. The table below shows the burden for three datasets/tasks. **The full curve is in the PDF**.
>
> | - | Average atom number | Average EOT computing time per sample |
> | --- | --- | --- |
> | QM9 | 18 | 1.10ms |
> | GEOM DRUG | 47 | 1.99ms |
> | Antibody-CDR | 150 | 18.84ms |
>
> Even for the Antibody-CDR data, with an average of 150 atoms, the process time is only 18.84 ms, which is acceptable in practice. Additionally, we can optimize the process further by leveraging its parallelizable nature.
> By Enabling prefetch and multiprocessing, we can minimize the computational overhead further and make it virtually inconsequential.
>
> ---
>
> ### Q2. The missing proof for the S(n)-invariant probability path of EOT.
>
> Thank you for bringing this to our attention. A detailed proof of proposition 4.4 can be found in Appendix B.3 in the submission, and we will include the reference in the updated version. Here we summarize the key steps of the proof and provide a simplified and intuitive explanation:
>
> The used dynamics model EGNN is permutation-invariant, and Zero CoM guarantees translation invariance. Thus, our goal is to demonstrate the invariance under rotations.
>
> For any rotation $\mathbf{T}$ on the point cloud $\mathbf{x}$ with $N$ points, the new $\mathbf{R}^*$ corresponding with $\mathbf{T}\mathbf{x}$ (we denote it by $\mathbf{R}^*_\text{rot}$) exactly offsets the impact of $\mathbf{T}$. Formally, let $\mathbf{x}_0$ denotes the target point cloud, we calculate
>
> $(\pi^*, \mathbf{R}^*)= \underset{(\pi, \mathbf{R})}{\operatorname{argmin}} ||\pi(\mathbf{R}\mathbf{x}^1, \mathbf{R}\mathbf{x}^2, \dots, \mathbf{R}\mathbf{x}^N) - \mathbf{x}_0||_2$, and
>
> $(\pi^*_\text{rot}, \mathbf{R}^*_\text{rot})= \underset{(\pi, \mathbf{R})}{\operatorname{argmin}} ||\pi(\mathbf{R}\mathbf{T}\mathbf{x}^1, \mathbf{R}\mathbf{T}\mathbf{x}^2, \dots, \mathbf{R}\mathbf{T}\mathbf{x}^N) - \mathbf{x}_0||_2$.
>
> Then $\pi^*_\text{rot}=\pi^*, \mathbf{R}^*_\text{rot} = \mathbf{R}^*\mathbf{T}^{-1}$. (Refer to Appendix B.3 for strict proof.)
>
> Since $p_1$ is invariant under rotations, and the transformation $\psi_t^{\mathrm{EOT}}$ satisfies
>
> $\psi_t^{\mathrm{EOT}}(\mathbf{x}) = (\sigma_{\min}+(1-\sigma_{\min})t)\pi^*(\mathbf{R}^*\mathbf{x})+(1-t)\mathbf{x}_0$, and
>
> $\psi_t^{\mathrm{EOT}}(\mathbf{T}\mathbf{x}) = (\sigma_{\min}+(1-\sigma_{\min})t)\pi^*_\text{rot}(\mathbf{R}^*_\text{rot}\mathbf{T}\mathbf{x})+(1-t)\mathbf{x}_0$.
>
> i.e. $\psi_t^{\mathrm{EOT}}$ is invariant.
>
> ---
>
> ### Q3. Can the authors provide some more motivation for using mutual information? Have the authors tried to scale one of the modalities to have the same variance as the other?
>
> 1. We explain the motivation for using mutual information. For atom types, the probability path determines when to predict the node type from timestep 0 (noise) to timestep 1 (data). In example 1 of line 214, one corner probability path has no noise from timestep 1 to timestep $1-\epsilon$ and maximum noise level from $1-\epsilon$ to timestep 0. When $\epsilon$ is small, the model tends to fix the node type at around ε step (early in the generation procedure), even if the coordinates are far from reasonable 3D structures. This makes it difficult for the model to learn or generate following such a path. The structure update in the following steps can change the bonded connection, causing a mismatch in valency with the early fixed atom types. Figure 4 shows that as the coordinates approach the ground truth, the predictability of atom types increases significantly. To align with this observation, we use mutual information as a guided bias to describe this tendency.
>
> 2. Yes, we have tried different inductive biases for matching the intermediate distributions including scaling different modalities to the same variance as the reviewer suggested. We provide the results for several matching strategies in the following table.
>
> | Metrics | Scaling the variance | Matching entropy in intermediate distribution | Matching Mutual Information(ours) | Without any matching |
> | ------------------- | -------------------- | --------------------------------------------- | --------------------------------- | -------------------- |
> | Mol Stability (QM9) | 81.3 | 83.5  | 88.3 | 77.1 |
>
> ---
>
> ### Q4. Missing related works:
>
> Thanks for introducing useful work to improve the paper. [1] and [2] propose using a joint distribution to replace independent coupling pairs between noise and data. Our method differs from previous research as follows: Previous work focuses on general domains like image generation, without exploring joint distribution design for complex geometries. Additionally, our EOT is conducted at the sample level, instead of batch level, allowing for parallelization.
> [3] advances 3D molecule generation by extending the equivariant diffusion model with a geometry latent space, orthogonal from our research. However, applying EquiFM to a similar latent space may be possible in the future. We will update references and discussions in the next version.
>
> [1] Pooladian et. al. Multisample Flow Matching: Straightening Flows with Minibatch Couplings, ICML 2023
>
> [2] Tong et. al. Conditional Flow Matching: Simulation-Free Dynamic Optimal Transport (Preprint)
>
> [3] Xu et. al. Geometric Latent Diffusion Models for 3D Molecule Generation. ICML 2023

---

> > ### Comment · Reviewer_zq9q · 2023-08-17
> >
> > I thank the authors for their answers!
> >
> > - I recommend adding Q2,3,4 to the revised version
> > - In you answer to Q1, the complexity of solving OT should be $O(n^3)$ correct? is it a typo in your answer? I would add this discussion as a limitation of the method, not scaling to very large molecules.
> >
> > Regarding other reviewer tN9X comments on clarity -
> > - I recommend adding citations where they could clarify what is done and to credit the previous works such as in section 3, mention in the beginning that you overview the Flow matching paper by Lipman et. al. and in section 4.1 when you mention the Zero-Com space, cite Hoogeboom et. al. etc..
> >
> >
> > I still believe this work has a novel contribution showing both the flexibility in FM framework and backing it up with strong empirical performance on known benchmarks.
> > I keep my score.

---

> > > ### Author Response · Authors · 2023-08-17
> > > **Thank you very much for your feedback!**
> > >
> > > Thank you so much for your valuable feedback and for recognizing our work!
> > >
> > > - We assure you that we will incorporate the discussion on Q2, Q3, and Q4 as suggested in the revised version of our paper.
> > >
> > > - Regarding Q1: you are right! Our method utilizes the scipy.optimize.linear_sum_assignment based on the Jonker-Volgenant algorithm, which indeed has a complexity of O(n^3). We will include a thorough discussion on the challenges faced when dealing with very large molecules to provide a more objective statement.
> > >
> > > - We are grateful for your suggestions on clarification. In the revised version, we will improve the citations to appropriately credit the previous works and ensure that our contribution is accurately described.
> > >
> > > If there is any additional information that you might need, please don't hesitate to inform us.

---

### Author Rebuttal · Authors · 2023-08-10

We would like to express our sincere appreciation to the reviewers and the area chairs for the efforts and time spent reviewing our work and the crucial, insightful, and constructive comments. Here we would like to highlight the main concerns of the reviewers  and the corresponding responses here:
1. **Scalability Issues**, we have provided extensive experiments and detailed analysis of the relationship between the computation overhead and molecule size. And we show that this does not leash the scalability of our methods.
2. **Evaluation Metrics**, we have added several evaluation metrics to better compare the chemical property of generated molecules.
3. **Ablation Study**, we have provided an extra ablation study for comparing the effect of different probability paths and the EOT maps.
4. **Presentation**, we will address all the typos and presentation issues as suggested by the reviewers.

---

### Decision · Program_Chairs · 2023-09-21

**Decision:**

Accept (poster)

**Comment:**

The paper adapts the Flow Matching framework for the problem of 3D molecule generation and is able to achieve state of the art performance on common benchmarks. The paper presents interesting contributions including incorporation of equivariance in Flow Matching, Optimal Transport alignment to straighten learned generation paths, and using different probability paths for different data modalities. The weaknesses include issues with the exposition, missing experiments/ablations, scalability to large molecules, and questions about computational overhead of the Hungarian algorithms. The authors already addressed some of these issues in the rebuttal and are requested to revise the paper accordingly in the camera ready version.